# Vertically grown ultrathin Bi$_2$SiO$_5$ as high-$\kappa$ single-crystalline gate dielectric

Jiabiao Chen [1], Zhaochao Liu[1], Xinyue Dong[1], Zhansheng Gao[1], Yuxuan Lin[1], Yuyu He[1], Yingnan Duan[1], Tonghuai Cheng[1], Zhengyang Zhou[2], Huixia Fu[3], Feng Luo[1] & Jinxiong Wu [1] ✉

Single-crystalline high-$\kappa$ dielectric materials are desired for the development of future two-dimensional (2D) electronic devices. However, curent 2D gate insulators still face challenges, such as insufficient dielectric constant and difficult to obtain free-standing and transferrable ultrathin films. Here, we demonstrate that ultrathin Bi$_2$SiO$_5$ crystals grown by chemical vapor deposition (CVD) can serve as excellent gate dielectric layers for 2D semiconductors, showing a high dielectric constant (>30) and large band gap (~3.8 eV). Unlike other 2D insulators synthesized via in-plane CVD on substrates, vertically grown Bi$_2$SiO$_5$ can be easily transferred onto other substrates by polymer-free mechanical pressing, which greatly facilitates its ideal van der Waals integration with few-layer MoS$_2$ as high-$\kappa$ dielectrics and screening layers. The Bi$_2$SiO$_5$ gated MoS$_2$ field-effect transistors exhibit an ignorable hysteresis (~3 mV) and low drain induced barrier lowering (~5 mV/V). Our work suggests vertically grown Bi$_2$SiO$_5$ nanoflakes as promising candidates to improve the performance of 2D electronic devices.

Two-dimensional (2D) semiconductors hold great promise for fabricating more-than-Moore transistors and exploring the emergent transport properties[1–5], but achieving their theoretical performance also requires compatible high-$k$ dielectrics to guarantee efficient gate control[6–10]. Compared to traditional amorphous dielectrics (Al$_2$O$_3$ and HfO$_2$)[11–13], single-crystalline gate insulators with dangling-bond-free surfaces are more competitive to fabricate high-performance 2D devices with reduced interfacial scatterings and gate hysteresis[14–18]. For example, hexagonal boron nitride (h-BN) has been widely used as a van der Waals (vdWs) substrate to improve carrier mobility and investigate exotic properties of 2D materials[19–23]. However, h-BN is also well known for its shortcomings of low dielectric constant ($k$ = 2-4) and harsh conditions (pressure: >4 G Pa, temperature: >1673 K) for the growth of high-quality single crystals[24]. Therefore, it is highly desirable to discover new vdWs insulators similar to h-BN, but has a much higher dielectric constant and more facile synthetic conditions, for exploring the emergent transport properties of 2D materials in a high-$\kappa$ dielectric environment, as well as for fabricating 2D-material-based electronic devices with scaled supply voltage[15,16]. Even so, very rare have succeeded[18,25]. Very recently, Peng et al.[18] reported the synthesis of Bi$_2$SeO$_5$ bulk crystals grown by chemical vapor transport (CVT), which can be exfoliated into few-nanometer-thick nanosheets and serve as an "h-BN" like high-$\kappa$ dielectric to improve the mobility of 2D materials and enable the observation of quantum Hall effects in Bi$_2$O$_2$Se. Nevertheless, the CVT process for the synthesis of Bi$_2$SeO$_5$ bulk crystals is also very time-consuming (typically ~40 days). Besides, similar to other vdWs dielectrics, transferable Bi$_2$SeO$_5$ nanoflakes with suitable thicknesses and domain sizes were also prepared by a low-efficiency method of mechanical exfoliation for subsequent vdWs device integration.

Compared to mechanical exfoliation, direct growth of free-standing ultrathin 2D insulators with high-$k$ nature by chemical

[1]Tianjin Key Lab for Rare Earth Materials and Applications, Center for Rare Earth and Inorganic Functional Materials, Smart Sensor Interdisciplinary Science Center, School of Materials Science and Engineering, Nankai University, Tianjin 300350, China. [2]State Key Laboratory of High Performance Ceramics and Superfine Microstructure, Shanghai Institute of Ceramics, Chinese Academy of Sciences, Shanghai 200093, China. [3]Center of Quantum Materials and Devices & College of Physics, Chongqing University, Chongqing 401331, China. ✉e-mail: jxwu@nankai.edu.cn

vapor deposition (CVD) is much more efficient, but remains challenging. Typically, CVD-grown atomically thin 2D insulators with a layered crystal structure preferably adopt an in-plane growth mode on substrates[26–29], which will inevitably set obstacles for clean sample transfer and subsequent vdWs integration. However, if an ultrathin 2D insulator can be vertically grown on the substrate, just like the case in $Bi_2O_2Se$[3,30,31], the transfer problem can be easily overcome due to much reduced interfacial interaction.

$Bi_2SiO_5$ is a well-known high-$\kappa$ dielectrics with an Aurivillius-type layered structure and a large band gap of 3.5-4.4 eV[32–35]. According to the previous works regarding the bulk single crystals and polycrystalline powders of $Bi_2SiO_5$, $Bi_2SiO_5$ shows an anisotropic dielectric constant[36,37] and its out-of-plane dielectric constant can be as high as 30-80[36–39], and thus was suggested as a potential candidate for high-temperature dielectrics[39]. Here, $Bi_2SiO_5$ was demonstrated as an excellent gate dielectric for 2D semiconductors. Ultrathin $Bi_2SiO_5$ single crystals with thickness down to monolayer were successfully synthesized by a facile CVD method, concurrently owing to the high dielectric constant (>30), large band gap (~3.8 eV), and large breakdown field strength. Remarkably, the preferable CVD growth mode of $Bi_2SiO_5$ can be regulated from in-plane to out-of-plane under optimized conditions, showing great feasibility on sample transfer by polymer-free mechanical pressing. Using ultrathin $Bi_2SiO_5$ nanoflakes as the vdW dielectrics and screening layers, we can greatly regulate the carrier density and improve the carrier mobility of few-layer $MoS_2$ (almost fifteen times higher than on the $SiO_2$ substrate at 5 K). Besides, the $MoS_2$ field-effect transistors (FETs) using $Bi_2SiO_5$ as dielectrics can operate at 0.5 V, exhibiting a large $I_{on}/I_{off}$ (>10^8), an ignorable hysteresis (~3 mV), low DIBL value (~5 mV/V) and low gate leakage current (~10^{-13} A).

## Results

### Structure, CVD growth, and characterization of layered $Bi_2SiO_5$

As shown in Fig. 1a, bismuth silicate $Bi_2SiO_5$ possesses a monoclinic lattice with $Cc$ space group (quasi-orthogonal, $a = 15.12$ Å, $b = 5.44$ Å, $c = 5.29$ Å, $\beta = 90.07°$) and has an Aurivillius-type layered crystal structure with alternatively stacked $[Bi_2O_2]_n^{2n+}$ and $[SiO_3]_n^{2n-}$ layers along the $a$-axis. The first-principle calculations were performed to investigate the band structure of layered $Bi_2SiO_5$. As shown in Fig. 1b, $Bi_2SiO_5$ exhibits a large direct band gap of ~3.79 eV, whose conduction band minimum (CBM) and valance band maximum (VBM) are both locate at Γ point of the first Brillouin zone and mainly originate from Bi-$p$ and O-$p$ orbits, respectively.

The essential prerequisite for using $Bi_2SiO_5$ as single-crystalline dielectrics with strong gate control is to achieve its growth of atomically thin films. However, the CVD growth of ultrathin $Bi_2SiO_5$ crystals remains a challenge yet. To our knowledge, $SiO_2$ is chemically inert and has a very high melting point, which prevents it to be used as Si supplier during the CVD growth of Si-based compounds to some extent. Nevertheless, it's well known that the fluorides will react with $SiO_2$ to form volatile Si-based precursors. As a result, we developed a facile CVD method for the synthesis of $Bi_2SiO_5$ ultrathin crystals by using the $BiF_3$ powders as Bi supplier and $SiO_2$ powders or quartz boat as Si supplier (see Supplementary Fig. 1). With this method, ultrathin $Bi_2SiO_5$ crystals with various thicknesses can be readily obtained on mica substrates. As shown in Fig. 1c, d and Supplementary Fig. 2, at a relatively high growth temperature (~1023 K), $Bi_2SiO_5$ adopts a preferable in-plane growth mode on mica, which is widely observed in the CVD synthesis of layered materials, revealing a transparent square-like shape and atomically thin nature (down to monolayer, Supplementary Fig. 3). Normally, the in-plane growth mode will result in a large attaching area and strong bonding force between the epitaxial layer and the substrate, thereby adding difficulties for sample transfer and subsequent vdWs integration. Remarkably, the preferable growth mode of $Bi_2SiO_5$ on mica can be regulated from in-plane to out-of-

plane growth modes just by lowering its growth temperature, which is similar to the case of $Bi_2O_2Se$[30]. As shown in Fig. 1e and Supplementary Fig. 2b, c, vertically grown ultrathin $Bi_2SiO_5$ crystals gradually emerge on mica at ~973 K, then dominates the CVD growth while further lowering the temperature to ~923 K. Unlike the case in Fig. 1c, vertically grown $Bi_2SiO_5$ crystals with thickness ranging from 7.5 to 50 nm can be easily transferred onto various substrates (such as $SiO_2$/Si, Fig. 1e) by a polymer-free mechanical pressing (Supplementary Fig. 4), showing an atomically flat surface even after sample transfer (Fig. 1f). The transferable feature without unfavorable residual polymer contaminations makes free-standing $Bi_2SiO_5$ appealing for fabricating vdWs heterojunction device. Occasionally, the CVD-grown $Bi_2SiO_5$ nanoflakes revealed a terraced morphology with a step height of ~0.76 nm (Fig. 1g), consistent with the theoretical value for the monolayer step in $Bi_2SiO_5$. It's worth noting that the CVD-grown $Bi_2SiO_5$ shows excellent air stability, whose surface morphology and roughness of $Bi_2SiO_5$ nanoplates remain almost the same when exposed to air for more than 7 months, which is an important metric for device fabrication as a high-$\kappa$ gate dielectric (Supplementary Figs. 5, 6). More details about the comparison experiments conducted to understand how the reaction goes in the CVD growth of $Bi_2SiO_5$ and the possible reason for its in-plane and out-of-plane growth can be found in the discussion part of supporting information (Supplementary Figs. 7, 8).

The crystalline phase of as-grown samples was confirmed by transmission electron microscopy (TEM), Raman spectroscopy, and X-ray diffraction (XRD). As shown in Fig. 1h, i and Supplementary Fig. 9, we performed high-resolution TEM imaging along three-zone axes [namely (010), (011), and (100)], as well as corresponding fast Fourier transform (FFT) fringes. Based on the atomic-resolved cross-sectional TEM technique, the alternative stacking of $[Bi_2O_2]_n^{2n+}$ and the $[SiO_3]_n^{2n-}$ layers with a layer space of 0.76 nm was observed. Besides, the averaged atomic ratio of Bi/Si was measured close to 2: 1 by energy dispersive spectroscopy (EDS, Supplementary Figs. 9, 10), which is consistent with the chemical formula of $Bi_2SiO_5$. The active modes at 70, 97, 149, 208, 298, 372, 433 cm⁻¹ of the Raman spectra also matched well with monoclinic $Cc$ phase of $Bi_2SiO_5$ (Supplementary Fig. 11)[35]. Additionally, the sharp XRD peaks, which can be assigned to ($h$00) crystal planes, further confirmed the high crystalline quality of the CVD-grown $Bi_2SiO_5$ nanoplates (Supplementary Fig. 11c).

### Dielectric properties of vertically grown $Bi_2SiO_5$ nanoplates

The transfer feasibility for vertically grown $Bi_2SiO_5$ nanoplates greatly facilitates the evaluation of their intrinsic properties, such as dielectric constant, band gap, and breakdown field strength. As shown in Fig. 2a, metal-insulator-metal (MIM) capacitors were fabricated on quartz substrates to extract the dielectric constant of $Bi_2SiO_5$ nanoflakes with varied thicknesses by capacitance-voltage ($C-V$) measurements. Here, the thick graphite and In/Au metals serve as the bottom and top electrodes, respectively. Thanks to high-$\kappa$ nature, $Bi_2SiO_5$ nanoflake with a thickness of 25.6 nm demonstrated a very high capacitance density of 1.12 μF/cm² at 100 Hz (Fig. 2a), revealing a slight decrease when the measuring frequency is up to 1 MHz. The corresponding capacitance-frequency ($C-f$) measurements showed the similar results (Fig. 2b). It is worth noting that the absolute capacitance value (>1 × 10⁻¹² F) measured is ~2-3 orders higher than the instrument's offset and noise level (<1.5 × 10⁻¹⁴ F, Supplementary Fig. 12). Based on the $C-V$ and $C-f$ data, we can estimate the effective permittivity ($\varepsilon_{eff}$) of $Bi_2SiO_5$ to be ~32.4 at 100 Hz, which is preferable than commercial amorphous high-$\kappa$ oxide such as $Al_2O_3$ ($k = 7$-9)[13,40,41] and $HfO_2$ ($k = 13$-17)[41,42], and higher than most of reported vdWs single-crystalline dielectrics, such as $h$-BN ($k = $~3.5)[43,44], $CaF_2$ ($k = 8.4$)[14], $Bi_2SeO_5$ ($k = 15.6$)[18], VOCl ($k = 11.7$)[45] and $ZrO_2$ ($k = 8$-19)[17]. Furthermore, we investigated the thickness-dependent $\varepsilon_{eff}$ by fabricating MIM capacitors with different $Bi_2SiO_5$ thicknesses, from which we can extract very large $\varepsilon_{eff}$ values of >30 in a wide thickness range (Fig. 2c).

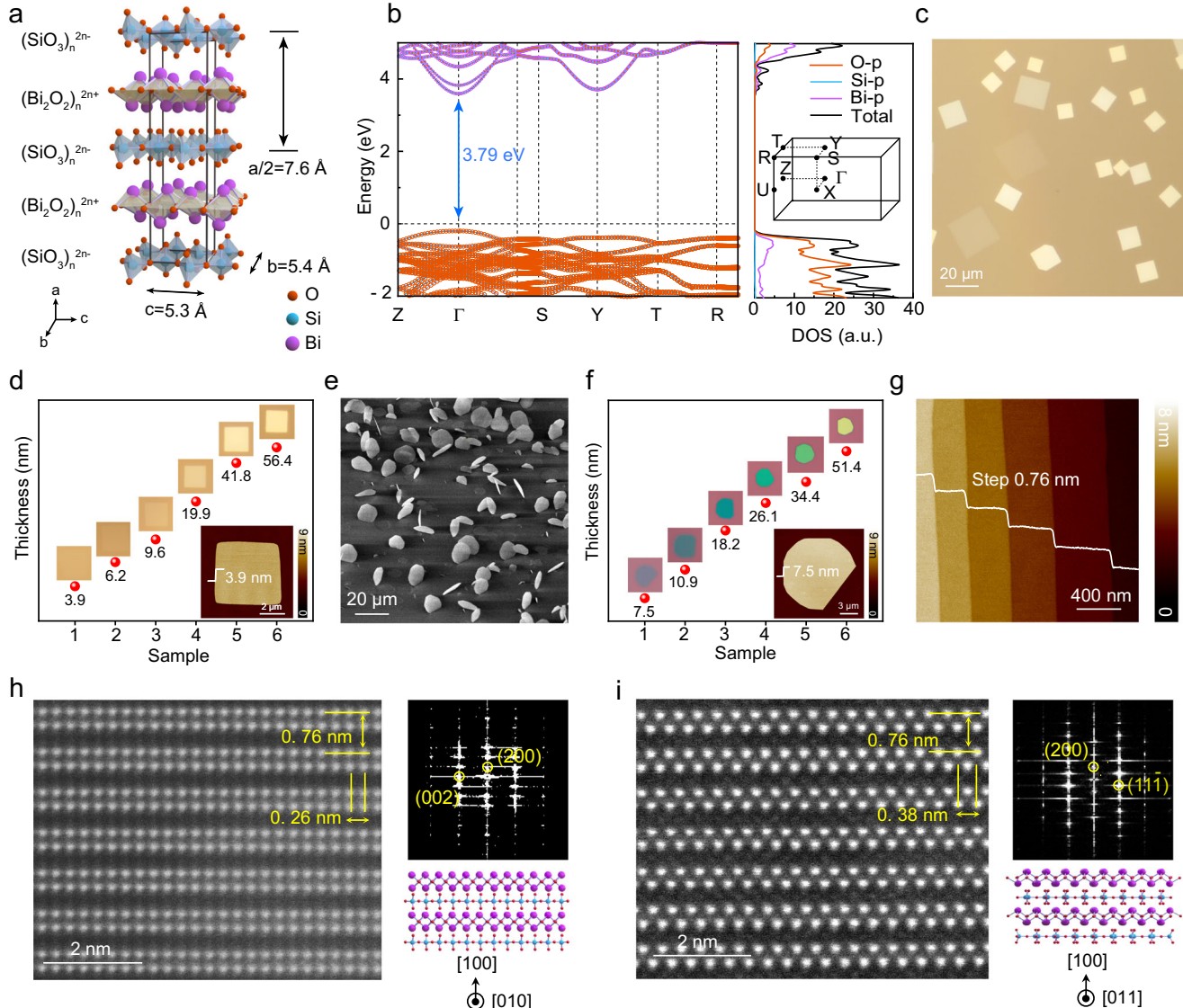

**Fig. 1 | Structure, growth, and characterization of ultrathin Bi₂SiO₅ single crystals. a** Crystal structure of Bi₂SiO₅ ($Cc$, $a = 15.12$ Å, $b = 5.44$ Å, $c = 5.29$ Å, $\beta = 90.07°$) with alternatively stacked $[Bi_2O_2]_n^{2n+}$ and $[SiO_3]_n^{2n-}$ layers. **b** Calculated band structure and density of states (DOS) of Bi₂SiO₅ with a direct band gap of ~3.79 eV. The first Brillouin zone is inserted in the right panel. **c** Typical optical micrograph (OM) image of square Bi₂SiO₅ nanoplates showing an in-plane growth mode on mica substrate. **d** OM images of Bi₂SiO₅ nanoplates with thickness-dependent color contrasts on mica. The inset shows the typical atomic force microscope (AFM) image of an ultrathin Bi₂SiO₅ nanoplate with a thickness of 3.9 nm (5 layers) and an atomically smooth surface. **e** Scanning electron microscopy (SEM) image of 2D Bi₂SiO₅ crystals vertically grown on mica substrate. **f** Thickness-dependent color contrasts for Bi₂SiO₅ nanoplates transferred onto SiO₂/Si substrate by a polymer-free mechanical pressing. The AFM image of a 7.5-nm-thick Bi₂SiO₅ nanoplate was inserted in **f. g** Typical AFM image of a terraced Bi₂SiO₅ nanoplates with a clear step of 0.76 nm. **h, i** Cross-sectional atomic-resolved high angle annular dark field (HAADF) images (left) and corresponding fast Fourier transform (FFT) diffraction spots (right) of chemical vapor deposition (CVD) grown Bi₂SiO₅ nanoplates taken along the zone axes of [010] (**h**) and [011] (**i**), respectively.

The gradual decrease of $\varepsilon_{eff}$ while thinning down can be ascribed to the existence of interfacial "dead layer" in the MIM device, which is similar to other MIM devices[15,46–48].

An ideal gate dielectric also needs a large band gap to inhibit the current leakage. Here, vertically grown Bi₂SiO₅ nanoflakes were directly transferred onto the polished quartz substrate by mechanical pressing for the ultraviolet-visible (UV-vis) absorption measurements. As shown in Fig. 2d, the optical band gap of CVD-grown Bi₂SiO₅ can be extracted as ~3.80 eV by Tauc's law, which is consistent with the theoretical value (3.79 eV, Fig. 1b). To illustrate the metrics of Bi₂SiO₅ as a gate dielectric, the relationship between band gap and dielectric constant of representative gate dielectrics in literature was plotted in Fig. 2e, clearly indicating the coexistence of high dielectric constant and large band gap in Bi₂SiO₅. Moreover, we evaluated the breakdown

field strength ($E_{bd}$) of the Bi₂SiO₅ nanoplates by conductive atomic force microscope (C-AFM) measurements on p⁺⁺ Si. As shown in Fig. 2f, the CVD-grown Bi₂SiO₅ showed thickness-dependent breakdown field strength ranging from 7.2 (13.5 nm) to 9.4 (8.2 nm) MV cm⁻¹, which is close to that of dielectrics in the silicon industry such as Al₂O₃[49] and HfO₂[50] but owns higher dielectric constant. We should emphasize that the C-AFM is a very local and microscopic tool to measure the breakdown voltage of a dielectric insulator, which may be not the same as the global one determined by MIM device. For example, the breakdown experiments based on the MIM devices gave a breakdown field strength of 3-5 MV/cm instead while varying the thickness of Bi₂SiO₅ from 10.1 to 21.4 nm (Supplementary Fig. 13).

The features of easy-to-transfer, ultra-flat surface, high dielectric constant, large band gap, and breakdown field strength make Bi₂SiO₅

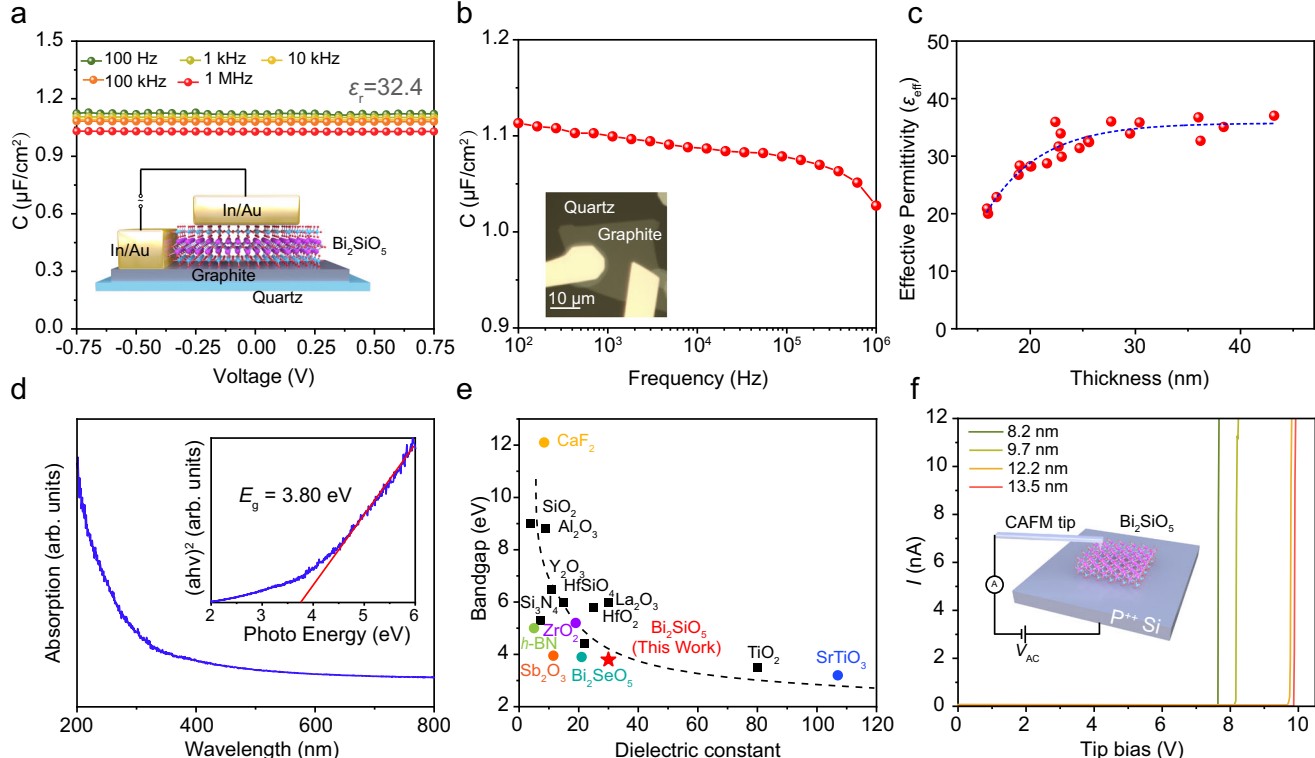

**Fig. 2 | Dielectric constant, band gap, and breakdown field strength of CVD-grown Bi₂SiO₅ nanoflakes. a** Typical Bias-dependent capacitance (*C*) measurements on CVD-grown Bi₂SiO₅ nanoflake with a common metal-insulator-metal (MIM) device configuration, where the thick graphite and In/Au metals serve as the bottom and top electrodes (inset), respectively. The dielectric constant ($\varepsilon_r$) of Bi₂SiO₅ was estimated as ~32.4 at 100 Hz. **b** Corresponding frequency-dependent capacitance (*C*–*f*) characteristics of the MIM device, whose OM image is inset in **b**. **c** The thickness-dependent effective permittivity ($\varepsilon_{eff}$) of Bi₂SiO₅ nanoflakes with a measuring frequency of 100 Hz. The dashed blue curve is a visual guide.

**d** Ultraviolet-visible (UV-vis) absorption spectrum of CVD-grown Bi₂SiO₅ nanoflakes transferred onto quartz substrate with high coverage by mechanical pressing. The inset shows the fitting of its optical band gap (~3.8 eV) by Tauc's law, where *a*, *h*, and *v* are the absorption coefficient, Planck constant and frequency, respectively. **e** Energy band gap versus dielectric constant of representative dielectric materials in literature, showing the coexistence of high dielectric constant and large band gap in Bi₂SiO₅. The dashed line is a visual guide. **f** Thickness-dependent current-voltage curves of Bi₂SiO₅ nanoplates measured by C-AFM, showing a high breakdown field strength of 9.4 MV/cm (8.2 nm).

highly competitive as gate dielectrics and high-$\kappa$ substrates for dielectric screening. Here, we combine Bi₂SiO₅ with mechanically exfoliated few-layer MoS₂ to construct Bi₂SiO₅/MoS₂ FETs for demonstrating its advantages as back-gate dielectrics (Fig. 3), dielectric screening substrate (Fig. 4) and top-gate dielectrics (Fig. 5).

**Vertically grown Bi₂SiO₅ as back-gate dielectric**
The small contact area and weak interaction between the vertically grown Bi₂SiO₅ and the mica substrate make it compatible with the well-developed aligned transfer method (For details, see Supplementary Fig. 14)[19,20], allowing us to fabricate complex vdWs heterojunction devices by layer-by-layer stacking. Figure 3a showed the OM image of an as-fabricated MoS₂ Hall-bar device using Bi₂SiO₅ as the back-gate dielectric and multi-layer graphene as a back-gate electrode. Gated Hall measurement is a powerful tool to get a series of key parameters, such as the dielectric constant of the gate dielectric, carrier density, and Hall mobility of the channel semiconductor. Besides, it also enables us to get the transfer and output characteristics by defining two out of six electrodes as source and drain. Figure 3b showed Hall resistance ($R_{xy}$) as a function of magnetic field (*B*) under different gate voltages ($V_g$) at room temperature, from which we can extract the $V_g$-dependent sheet carrier densities ($n_{2D}$). By linear fitting the curve of $n_{2D}$-$V_g$ (Fig. 3c), the estimated slope equals the $C_{ox}/e$, where $C_{ox}$ is capacitance density and e is the elementary charge. The $C_{ox}$ for a 22-nm-thick Bi₂SiO₅ can be as high as 1.45 μF/cm², suggesting a very high dielectric constant ($\varepsilon_r$ ~ 36), which is consistent with the value extracted by *C*–*V* measurements in a reasonable accuracy. On the other hand, the

very large capacitance density suggests that we can greatly regulate the carrier density and electrical behavior of the MoS₂ semiconductor. As confirmed by the room-temperature Hall measurements (Fig. 3c), a very high carrier density of $1.83 \times 10^{13}\,\text{cm}^{-2}$ can be doped into the channel by applying a $V_g$ of 2 V on Bi₂SiO₅ dielectrics. To this end, the temperature-dependent longitudinal resistance ($R_{xx}$-*T*) behavior of MoS₂ can be greatly regulated from insulating to metallic one when sweeping the $V_g$ from 0.4 to 2 V (Fig. 3d), indicating the great potential of Bi₂SiO₅ dielectric as a powerful tool for physical state regulation.

To examine whether ultrathin Bi₂SiO₅ can serve as excellent back-gate dielectrics of a FET, we define the electrodes #1, #4 and graphene as source, drain, and gate electrodes, respectively. The dual-sweep transfer curves of MoS₂/Bi₂SiO₅/Gr FET were presented in Fig. 3e, showing a large $I_{on}/I_{off}$ of >$10^8$, and a small SS value of ~64 mV/decade. Considering the relatively complex device fabrication process (Supplementary Fig. 14), chemical residuals will inevitably remain at the interfaces between the graphene, Bi₂SiO₅ and MoS₂, which are detrimental to the device performance. Nevertheless, the FET still exhibited small hysteresis of ~15 mV, which is comparable to the value obtained after interface optimization[51]. Thanks to the Ohmic contact formed by In/Au electrodes (Fig. 3f), two-terminal field-effect mobility of the device can be as high as 18.6 cm² V⁻¹ s⁻¹ by linear fitting the transfer curve, which is comparable to the Hall mobility measured at room temperature (23.7 cm² V⁻¹ s⁻¹ at $V_g = 2$ V, Supplementary Fig. 15). Notably, Bi₂SiO₅ was also verified as an excellent gate insulator while shrinking the channel length of MoS₂ FETs down to 100 nm and even 30 nm (Supplementary Figs. 16, 17).

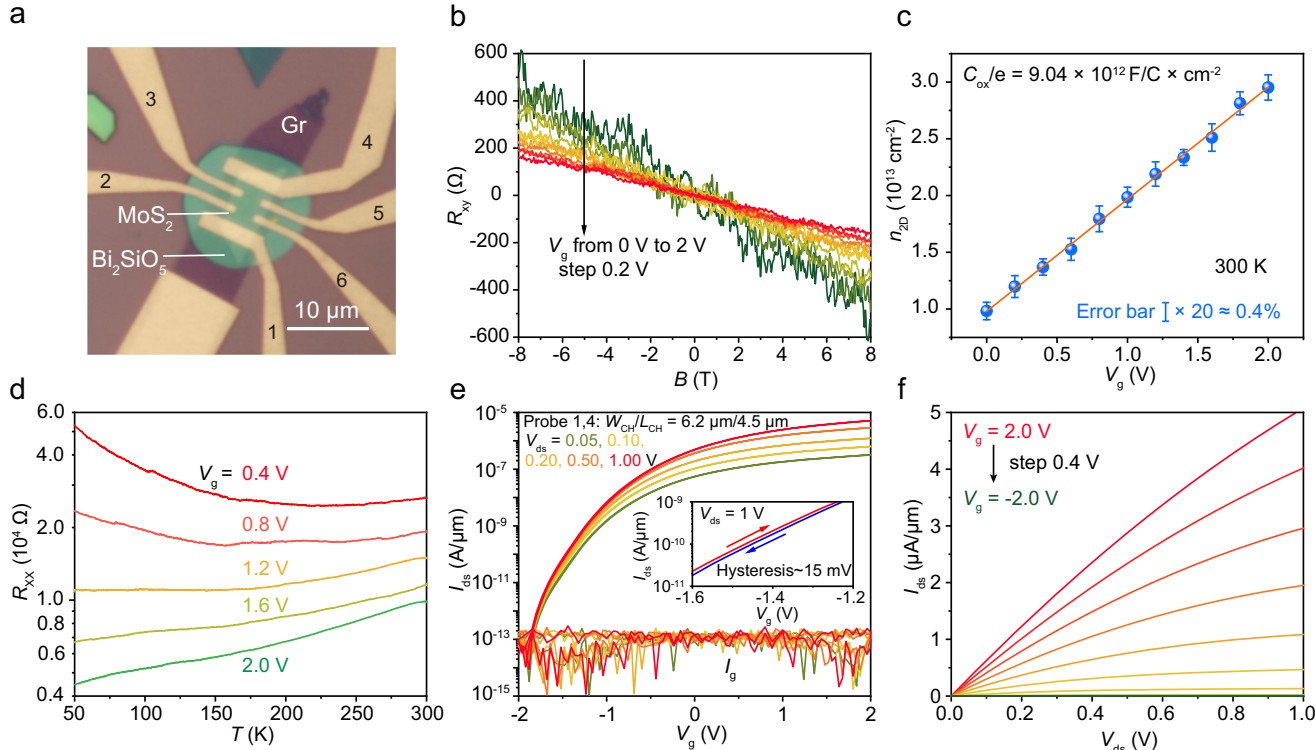

**Fig. 3 | MoS₂ Hall-bar device with the vertically grown ultrathin Bi₂SiO₅ nanoflake as the high-κ back-gate dielectrics. a** OM image of an as-fabricated MoS₂ (~2.7 nm) Hall-bar device on SiOₓ/Si substrate, in which a thin flake of Bi₂SiO₅ (~22 nm) and multi-layer graphene were adopted as gate dielectric and back-gate electrode, respectively. **b** Hall resistance ($R_{xy}$) as a function of magnetic field ($B$) under various gate voltages ($V_g$) from 0 to 2 V at 300 K. **c** The extracted $V_g$-dependent sheet carrier density ($n_{2D}$), in which a high dielectric constant ($\varepsilon_r$) of ~36 was derived by linear fitting. The Error bars are based on the standard deviations of slope fitting on $R_{xy}$-$B$ curves. **d** Longitudinal resistance ($R_{xx}$) as a function of temperature ($T$) under different $V_g$ from 0.4 to 2.0 V, showing a gate-induced insulator-to-metal transition. **e** Two-probe transfer curves of the back-gated MoS₂/Bi₂SiO₅ field-effect transistor (FET) measured by defining the probes of 1 and 4 as source and drain terminals, showing $I_{on}/I_{off} > 10^8$, $SS$ ~ 64 mV/decade. The inset shows a small hysteresis of ~15 mV. The extracted two-terminal field-effect mobility of the device is 18.6 cm² V⁻¹ s⁻¹. **f** The corresponding output curves measured in a $V_g$ range from 2 to −2 V.

## Vertically grown Bi₂SiO₅ as high-κ screening layer

Dangling bonds on SiO₂/Si substrate usually act as the scattering sites of Coulomb impurities (CI) to decrease the mobility of a semiconductor[52–54]. Theoretically speaking, using an *h*-BN-like high-κ substrate free of dangling bonds can effectively enhance its mobility due to reduced CI scatterings and excellent dielectric screening. Figure 4a shows the scheme and OM image of as-fabricated 4-probe MoS₂ FETs on top of Bi₂SiO₅ and SiO₂/Si substrates. To avoid the influence of sample quality variation, one MoS₂ sample (4.3 nm) was simultaneously placed on Bi₂SiO₅ (31.9 nm) and SiO₂ substrates for comparison. As demonstrated in Fig. 4b, the $V_g$-dependent two-probe I–V curves (output characteristics) were measured at 300 K, showing a linear behavior over a large voltage window, confirming the Ohmic contact formed by In/Au electrodes. It's worth noting that the on-state current ($I_{on}$) of MoS₂ on Bi₂SiO₅ is about twice as the one on SiO₂, suggesting a higher mobility of MoS₂ on Bi₂SiO₅ substrate. To illuminate the influence of contact resistance, 4-probe transfer curves were measured among a temperature range of 5-300 K (Fig. 4c). By linear fitting the transfer curves, we can extract the 4-probe FET mobility ($\mu_{FET,4\text{-}probe}$) as a function of temperature (Fig. 4d). Among the whole temperature range of 300-5 K, the carrier mobility of MoS₂ on Bi₂SiO₅ is significantly higher than that on SiO₂/Si substrate, which can be further confirmed by plotting the $I_{ds}$ as a function of a normalized $V_g$ by threshold voltage ($V_g$-$V_{th}$, Supplementary Fig. 18). Particularly, the mobility of MoS₂ on Bi₂SiO₅ at 5 K is as high as 549.3 cm² V⁻¹ s⁻¹, which is almost fifteen times higher than the value of MoS₂ on SiO₂/Si (~37.4 cm² V⁻¹ s⁻¹). Moreover, the carrier mobility of MoS₂ on Bi₂SiO₅ and SiO₂/Si showed totally different temperature dependence. For

MoS₂/Bi₂SiO₅, its carrier mobility increased monotonously upon cooling down, suggesting the phonon scattering dominated the whole transport events even at low temperature. In contrast, the carrier mobility of MoS₂ on SiO₂/Si substrate increased first while cooling down to ~150 K, but gradually decreased upon further cooling. This $T$-dependent mobility can be well explained by enhanced charge impurities scattering at low temperature in MoS₂/SiO₂ interface. In a word, using Bi₂SiO₅ as a substrate greatly improve the performance of the MoS₂ FET to get a higher mobility, which can be attributed to the suppressed CI scatterings in the high-κ surroundings and ideal dielectric/semiconductor interface.

## Vertically grown Bi₂SiO₅ as the top-gate dielectric

Top-gate FET is a widely used device configuration in practical applications. In this part, we fabricated Bi₂SiO₅/MoS₂ top-gate FETs to prove the feasibility of Bi₂SiO₅ as top-gate dielectric layer in hysteresis-free low-power transistors (Supplementary Fig. 19). As shown in Fig. 5a, the Bi₂SiO₅/MoS₂ top-gate FET was placed on the SiO₂/Si substrate, which can also serve as the back-gate dielectric and electrode when necessary. Figure 5b showed the typical dual-sweep transfer curves of a 22.9-nm-thick Bi₂SiO₅-based MoS₂ FET measured under different $V_{ds}$ from 0.05 to 1 V. Owing to the excellent gate tunability, the transistor can be effectively turned on and off by applying a $V_g$ within the range of −0.6-0.5 V, showing a large $I_{on}/I_{off}$ of >10⁶ and a low gate leakage current of 10⁻¹³ A (approaching the detection limit). Furthermore, the ideal SS value (~62 mV/decade), ignorable gate hysteresis (~3 mV), and low drain-induced barrier lowering (DIBL, ~5 mV/V) suggest the perfect interface was formed at MoS₂/Bi₂SiO₅ interface (Supplementary

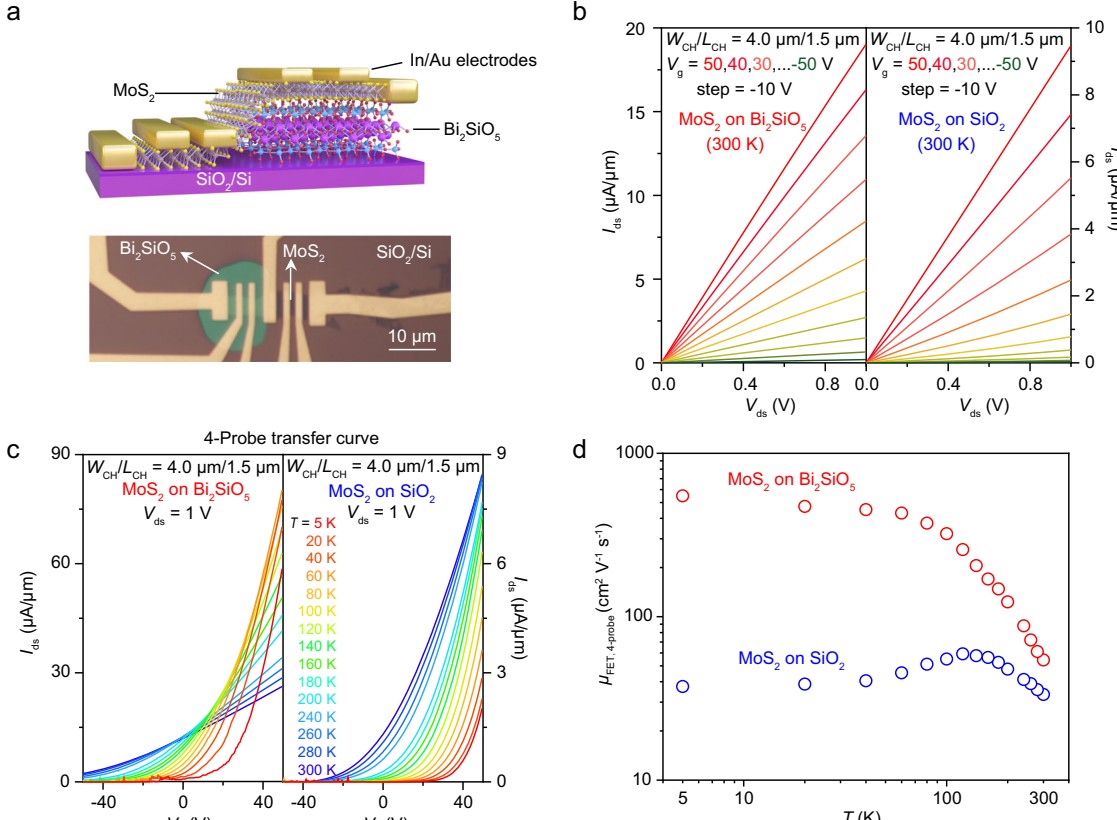

**Fig. 4 | Dielectric screening and mobility enhancement effects of CVD-grown $Bi_2SiO_5$ nanoflakes as the high-$\kappa$ substrates. a** Schematic illustration and OM image of back-gate $MoS_2$ four-probe FETs device. **b** Linear output curves ($I_{ds}$–$V_{ds}$) of $MoS_2$ FETs on $Bi_2SiO_5$ (left) and on $SiO_2$ substrate (right) measured at 300 K. **c** The 4-probe transfer curves of $MoS_2$ FETs measured at different temperatures (5-300 K) on $Bi_2SiO_5$ (left) and $SiO_2$ (right) substrates, respectively. **d** The extracted temperature-dependent 4-probe FET mobility ($\mu_{FET,4\text{-probe}}$) of $MoS_2$ on $Bi_2SiO_5$ (red) and $SiO_2$ (blue) substrates.

Fig. 20). The $MoS_2$ transistor showed a linear $I_{ds}$-$V_{ds}$ curve at low $V_{ds}$ region, then gradually saturated at high $V_{ds}$ region (Fig. 5c). Particularly, the SS remains low (<70 mV/decade) for different $I_{ds}$ of several orders of magnitude for both forward and reverse top-gate sweeping (Fig. 5d). More importantly, the top-gate $Bi_2SiO_5$/$MoS_2$ short-channel FET, whose channel length was defined by the gap distance between two graphene electrodes (~180 nm), still showed a small DIBL value of ~22 mV/V and SS value of ~79 mV/decade (Supplementary Fig. 21).

One step further, nearly hysteresis-free transfer curves were preserved in a dual-gate FET configuration, in which a back-gate voltage ($V_{BG}$) was employed to modulate the threshold voltage ($V_{th}$) of the device. As a result, the top-gate transfer curves gradually shifted as the $V_{BG}$ varied from 5 to 0 V (Fig. 5e and Supplementary Fig. 22). By linear fitting the $V_{th}$ as a function of $V_{BG}$, we can extract a slope of −0.0097, which equals to the ratio of the bottom-gate to top-gate capacitance, namely $C(SiO_2)/C(Bi_2SiO_5)$, when the parallel-plate capacitor model is assumed for both top and bottom gates[16,55]. For a 285 nm $SiO_2$ dielectric ($\varepsilon_r$ = 3.9), its capacitance can be calculated as 0.0121 μF/cm². In this case, the capacitance and dielectric constant for a 22.9-nm-thick $Bi_2SiO_5$ were derived as 1.25 μF/cm² and 32.3, both of which matched well with the value obtained by $C$–$V$ (Fig. 2a) and gated Hall measurements (Fig. 3c). Next, the interface trap density $D_{it}$ was extracted based on the following equation:[15]

$$SS = \ln(10)\frac{k_B T}{q}\left(1 + \frac{qD_{it}}{C_{ox}}\right) \qquad (1)$$

where $SS$ is the subthreshold swing, $k_B$ is Boltzmann constant, $T$ is absolute temperature, $q$ is the elementary charge, $C_{ox}$ is the gate

capacitance obtained from MOS capacitance measurements. As a result, a low $D_{it}$ value of $2.88 \times 10^{11}$ cm⁻²/eV was extracted, verifying the high quality of the vdWs interface.

Moreover, the dielectric constant of $Bi_2SiO_5$, extracted by dual-gate transfer measurements, also showed similar thickness dependence with the $C$–$V$ results (Fig. 5g). Figure 5h plotted the dual-sweep transfer curves of $MoS_2$ FETs with different $Bi_2SiO_5$ thicknesses (10.1-67.5 nm). Apparently, smaller $V_g$ is needed to switch the transistor on and off when a thinner $Bi_2SiO_5$ dielectric is used. We should emphasize that the EOT value for a 10.1 nm $Bi_2SiO_5$ is as small as 1.3 nm, but its gate leakage current is still on the order of $10^{-13}$ A, signifying substantial room space for further scaling of EOT and great potential applications in low-power devices. The $I_{on}$ of $MoS_2$ FETs in Fig. 5h seems to decrease with decreasing the thickness of $Bi_2SiO_5$, which may originate from the contact issues existing in the $MoS_2$ FETs with the thin $Bi_2SiO_5$ as gate insulators. However, we should emphasize that, the $I_{on}$ of ultrathin $Bi_2SiO_5$-gated $MoS_2$ FETs can be greatly improved by optimizing the device fabrication process. As shown in Supplementary Figs. S23–25, the $I_{on}$ of another 10-nm-thick $Bi_2SiO_5$-gated $MoS_2$ can be similar to the value of the thick $Bi_2SiO_5$-gated FETs (0.11 μA/μm, Fig. 5b). It is worth noting that the $I_{on}$ is greatly limited by the remaining ungated channel in the top-gate device configuration. Additionally, as confirmed by the dual-sweep transfer curves, a nearly ideal SS value and a low normalized gate hysteresis can indeed be obtained in the $Bi_2SiO_5$ gated $MoS_2$ FET.

The low operating voltage is essential to fabricate low-power logic circuits. As demonstrated in Fig. 5i, we used two n-type transistors as the load and driver terminal to construct 2D inverter with a high voltage gain of 22.0. The inverter can demonstrate the logic state 0 and 1

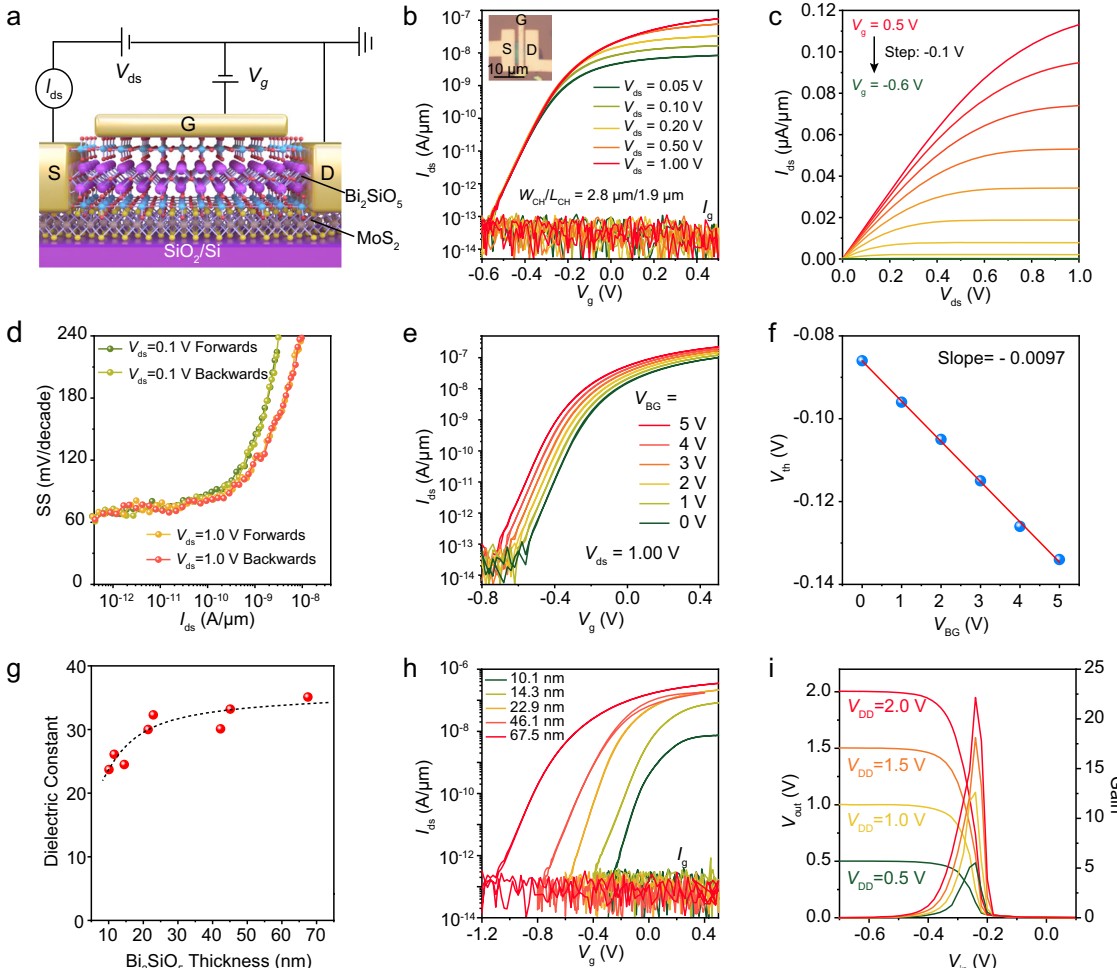

**Fig. 5 | Hysteresis-free MoS₂ FETs and low-power NMOS inverters using Bi₂SiO₅ as high-κ top-gate dielectrics. a** Schematic illustration of the top-gated MoS₂ FETs on SiO₂/Si substrate with Bi₂SiO₅ as the top-gate dielectric. **b** Typical dual-sweep transfer curves of the MoS₂/Bi₂SiO₅ FET measured under different $V_{ds}$ from 0.05 to 1 V, showing an ideal *SS* value of ~62 mV/decade and ignorable gate hysteresis. The Insert is the OM image of a fabricated MoS₂ FET with Bi₂SiO₅ as the gate dielectric. **c** Corresponding output characteristics ($I_{ds}$–$V_{ds}$ curves) of the device measured by varying the $V_g$ from 0.5 to −0.6 V with a 0.1 V step. **d** Extracted *SS* value versus $I_{ds}$ characteristics of the device in **b**, showing a low SS value (<70 mV/decade) for a wide $I_{ds}$ range. **e** Dual gated transfer curves of the MoS₂/ Bi₂SiO₅ FET under different

back-gate voltages ($V_{BG}$) from 5 to 0 V. **f** The extracted threshold voltage $V_{th}$ from **e** as a function of $V_{BG}$. The linear fitting yields a slope of −0.0097 and a high dielectric constant of ~32.3. **g** The thickness-dependent dielectric constant of Bi₂SiO₅ extracted by dual-gate measurement on Bi₂SiO₅/MoS₂/SiO₂/Si FET. The dotted curve is a visual guide. **h** Transfer curves of the MoS₂ FETs with different thickness Bi₂SiO₅ as the top-gate dielectrics, showing a trend of smaller $V_{th}$ for thinner Bi₂SiO₅ thickness. The $V_{ds}$ is 0.5 V, and no $V_{BG}$ is applied. **i** Measured output voltage ($V_{out}$) and gain as a function of input voltage ($V_{in}$) of an NMOS inverter based on two MoS₂/ Bi₂SiO₅ FETs under different supply voltage ($V_{dd}$) from 2 V to 0.5 V.

within ±0.5 V, showing a low dynamic power consumption of <0.7 nW (Supplementary Fig. 26).

In summary, our work achieved the direct CVD growth of ultrathin free-standing high-*k* single-crystalline dielectrics, which is much more efficient than traditional mechanical exfoliation. The vertically grown Bi₂SiO₅ 2D crystals present the metrics for a gate dielectric, as evidenced by the coexistence of high dielectric constant, large band gap, high breakdown field strength, as well as the characteristic of easy transfer by facile polymer-free mechanical pressing. These features make Bi₂SiO₅ attractive as an inert vdWs substrate superior to *h*-BN for exploring exotic transport properties under reduced interfacial scatterings and stronger gate control, as well as for fabricating hysteresis-free 2D transistors with scaled supply voltage.

## Methods
### CVD growth of ultrathin Bi₂SiO₅ nanoplates on mica substrate
2D Bi₂SiO₅ crystals were synthesized inside a homemade CVD system equipped with a single heating zone tube furnace and 30 mm diameter quartz tube. Typically, the BiF₃ powders (purity 99.999%, Macklin)

were placed in an empty quartz boat or on top of the SiO₂ powders located in the heating center, and the freshly cleaved fluorophlogopite mica substrates were placed above the quartz boat. The heating temperature of the source was 600–750 °C and the growth time was 20 min. The 50 s.c.c.m Ar and 5 s.c.c.m mixed Ar/O₂ (1‰) gas were introduced into the CVD system as carrier gas. The system pressure was kept constant as 760 Torr during the whole growth process.

### Characterization of CVD-grown ultrathin Bi₂SiO₅ single-crystalline dielectric
The morphologies of as-synthesized in-plane and free-standing 2D Bi₂SiO₅ nanoplates were characterized by optical microscopy (Olympus BX53), scanning electron microscopy (SEM, JSM-7800F) and atomic force microscope (AFM, Bruker dimension icon). With a polymer-free method of mechanical pressing, vertically grown Bi₂SiO₅ nanoflakes were transferred onto Cu grid, glass, and optical quartz substrates to perform the characterizations of transmission electron microscopy (TEM, JEM 2800), X-ray diffraction (XRD, Rigaku Smart Lab 30 KW) and absorption spectrum (SHIMADZU, UV-2600),

respectively. The Raman spectroscopy was measured on WITec alpha300R with a laser of 532 nm. The cross-sectional TEM samples of in-plane and vertically grown $Bi_2SiO_5$ nanoflakes were both prepared by using a focused ion/electron dual beam system (FEI, Helios 5 CX). All cross-sectional scanning transmission electron microscopy (STEM) imaging was obtained on an aberration-corrected TEM operating at 300 kV (FEI Titan cubed Themis G2 300). The breakdown field strength of $Bi_2SiO_5$ was measured with the C-AFM module of Bruker Dimension Icon.

### First-principles calculations

The structural relaxation of $Bi_2SiO_5$ is performed within the framework of density function theory (DFT) using the projector augmented wave pseudopotential and the Perdew–Burke–Ernzerhof exchange-correlation functional as implemented in the VASP. The energy cut-off for the plane-wave expansion is set to 500 eV, and a Monkhorst–Pack k-mesh of $3 \times 9 \times 9$ is used in the Brillouin zone. The energy convergence threshold is $10^{-6}$ eV and the force $10^{-3}$ eV Å$^{-1}$ in the structural optimization. In order to overcome the underestimation of energy gap from the generalized gradient approximation (GGA), we use the method of modified Becke–Johnson potential (mBJ) to calculate the electronic structure.

### Device fabrication

To illuminate the possible capacitance coupling, the $Bi_2SiO_5$-based metal-insulator-metal (MIM) capacitors were fabricated on quartz substrates rather than $SiO_2/Si$ substrates. First, the thick graphite was exfoliated onto the quartz substrate as bottom electrodes for its ultrasmooth surface. Next, the vertically grown $Bi_2SiO_5$ nanoflakes were directly picked up from the mica substrate by a polypropylene carbonate/polydimethylsiloxane (PPC/PDMS) stamp, followed by aligned transfer onto the specific graphite bottom electrode by a high-precision transfer platform. Subsequently, the standard electron-beam lithography (EBL) process and thermal evaporation were used to pattern the top electrodes ((In/Au, 5/40 nm).

For fabricating the back-gated $MoS_2$ Hall-bar device using graphite as bottom electrodes, the detailed process was listed as follows. First, $SiO_2/Si$ substrates (285 nm $SiO_2$) were pretreated with $O_2$ plasma (Diener Pico plasma cleaner) for 5 min at a power of 50 W. Next, few-layer $MoS_2$ and graphite were exfoliated onto different $SiO_2/Si$ substrates. With the help of the PPC/PDMS stamp and high-precision transfer platform, the free-standing $Bi_2SiO_5$ and $MoS_2$ nanoflakes were sequentially stacked on top of the graphite. The six-terminal electrode legs for the Hall-bar and the bottom metal electrodes were simultaneously written by one-step EBL and following thermal metal deposition (In/Au, 5/40 nm). The device fabrication process of 4-terminal $MoS_2$ FET includes the following parts: (1) transfer the $Bi_2SiO_5$ onto $SiO_2/Si$ substrate by mechanical pressing; (2) place part of the few-layer $MoS_2$ on top of the $Bi_2SiO_5$ nanosheet; (3) EBL and metal deposition (In/Au, 5/40 nm).

For the top-gated $MoS_2$ device, few-layer $MoS_2$ nanosheets were exfoliated onto $SiO_2/Si$ substrate, followed by stacking $Bi_2SiO_5$ nanoflakes in the middle of the $MoS_2$ nanosheets as the gate dielectrics. The source, drain, and top-gate electrodes of $MoS_2$ FETs were patterned together with one-step EBL process and thermal metal evaporation (In/Au, 5/40 nm).

### Electrical transport measurements

2-probe electrical properties of the back-gate and top-gate $MoS_2$ FETs, including the Figs. 3e, f, 4b, 5b, c, e, h, were carried out by a semiconductor analyzer (FS-Pro) in a shielded vacuum chamber (<0.1 Torr) at room temperature, whose noise level is -1 × 10$^{-13}$ A within the voltage range of ±2 V. The 4-probe transfer curves and gated 4-probe measurements (such as gated $R_{xx}$-$T$ and Hall data), including Figs. 3b, d, 4c, were carried out in a Physical Properties Measurement Systems (PPMS-

9T, Quantum Design) equipped with a homemade electrical measurement system, which is composed of 2 Keithley 2400, and 2 Keithley 2182 A nanovoltmeter and has a noise level of -10$^{-10}$ A within the voltage range of ± 2 V. The $C-V$ and $C-f$ measurements were carried out on a FS336 LCR Meter.

## Data availability

Relevant data supporting the key findings of this study are available within the article and the Supplementary Information file. All raw data generated during the current study are available from the corresponding authors upon request.

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

## Acknowledgements

We thank professor Hailin Peng @ Peking University, professor Keji Lai and Shaopeng Feng @ UT-Austin for constructive discussions and suggestions, and acknowledge financial support from the National Natural Science Foundation of China (no. 92064005)) and the Opening Project of State Key Laboratory of High-Performance Ceramics and Superfine Microstructure (SKL202211SIC).

## Author contributions

J.W. convinced the original ideas and supervised the whole project. J.C. performed the CVD growth, characterization, device fabrication, and electrical measurements under the assistance of Z.L., Y.L., Y.H., Y.D., and T.C. Z.G. assisted by Z.Z. performed part of the TEM characterizations. Z.L. plotted the cartoon diagram. X.D. performed the theoretical calculation under the supervision of H.F. J.W. and J.C. wrote the paper with the input of other authors. F.L. co-supervised the whole project and gave constructive suggestions. All authors contributed to the scientific discussions.

## Competing interests

The authors declare no competing interest.
