## [Peer Review File · Nature Communications]

Vertically grown ultrathin insulator as high- κ single-crystalline gate dielectricREVIEWER COMMENTS

Reviewer #1 (Remarks to the Author):

In this study, Chen et al reported high-quality ultrahigh-k Bi₂SiO₅ and constructed high-performance devices. Overall, I think the work could be impactful. But, I am not able to recommend a publication, unless the authors address the following issue.

1.

Bi₂SiO₅ is the central material of the study. Hence, it deserves a thorough introduction on its advantages and uniqueness, especially its theoretical and existing experimental results on dielectric constant, bandgap, and dielectric strength, to justify the application of this material.

2.

All figures are very blurry, and most of the font sizes are very small. Many labels and values are impossible to read.

3.

Figure 3b is noisy. Hence, the corresponding calculations of carrier density and dielectric constant in Fig. 3c should have a noticeable error bar and variation, respectively.

4.

The author should present calibration on and raw data of capacitance measurement. In consideration of the small size, the capacitance measurement presented in Fig. 2 is generally very challenging and often produce large variations and errors. Hence, in order to reliably present the results, the author needs to present their protocol of offset calibration of the equipment, calibration of equipment noise level, effective area estimation, and dielectric constant calculation, at least in SI.

5.

The breakdown voltage measured by AFM is local and not the same as the global one. The author shall admit to avoiding confusing the readers.

6.

For the FET measurement, if the SS is 62 mV/decade, the author cannot label 60 mV/decade in the figure. This will confuse the readers.

7.

The author was not clear about which measurement was done using Keithley 2400 and PPMS. What equipment and type of cable did the author use to measure some of the figures, such as Figure 3b, 3e, 4c, 5b, 5e, and 5h? Keithley2400 and PPMS cable generally have high noise levels.

8.

Did the author normalize the curves or shift the curves vertically or horizontally?

Reviewer #2 (Remarks to the Author):

In this work, the authors report that ultrathin Bi₂SiO₅ crystals (down to monolayer) grown by chemical vapor deposition (CVD) can serve as an excellent gate dielectric for 2D semiconductors, showing an ultrahigh dielectric constant (>30), large band gap (~3.8 eV) and large breakdown field strength (9.4 MV/cm). Although the material is novel, the results of application in 2D dielectrics are far from satisfactory. I don't recommend to publish it in Nat. Comm.

1. It is well known that band offsets between semiconductors and dielectrics are important and necessary parameters in material and device design. For practical use, both the valence and conduction band offset (VBO and CBO) should be sufficiently large (> 1.0 eV) to block tunneling of electrons or holes from semiconductors. The bandgap of dielectrics usually needs to be larger than 5 eV. Therefore, claiming wide-gap insulator is not correct.
2. There is a trade-off between K value and the bandgap condition, which requires a reasonably large band gap [Materials Science and Engineering R 88 (2015) 1–41]. Compared to the common dielectrics, the bandgap of this material is narrow, but its breakdown field is comparable to that of SiO₂ (the bandgap of SiO₂ is 9 eV). What is the mechanism behind it? The C-AFM measurement is reliable? In semiconductor area, MIM structure are usually used to evaluate the breakdown field.
3. For dielectrics in 2D materials, the key parameters of transistors are mobility, drain current of the short channel, and EOT scaling. This paper did not focus on these important device parameters, which need to be improved to determine the potential of this material compared to BN and Bi₂O₅Se.
4. All the drain currents in this work should be normalized by channel width.
5. In fig. 4, the MoS₂ devices on Bi₂SiO₅ and SiO₂ have different threshold voltages, indicating different carrier densities in fig. 4b. Therefore, to make a fair comparison, using $V_g - V_{th}$ is recommended.
6. In Fig. 5f, the shift of V_{th} is just ~0.05 V. However, in fig. 5e, the shift is actually ~0.2 V by changing V_{bg} if $I_{ds} = 0.1$ nA. Therefore, the slope is problematic.
7. In Fig. 5h, the performance of MoS₂ transistor decreases with decreasing the thickness of BiSiO. What is the reason? The V_g range is too small. Pushing it to 2 V is reasonable in view of high breakdown field of BiSiO.
8. To evaluate the interface trap, the interface trap density D_{it} is a quantified parameter. Please provide this information.

Reviewer #3 (Remarks to the Author):

Chen et al. achieved successful growth of ultrathin Bi₂SiO₅ crystals using chemical vapor deposition (CVD). These crystals possess advantageous properties, including a high dielectric constant, a large band gap, and high breakdown field strength. An important feature of the few-layer Bi₂SiO₅ dielectrics is their vertical growth on a mica substrate with weak interfacial interaction. This allows for easy transfer onto other substrates through mechanical pressing, facilitating ideal van der Waals integration with other 2D materials like few-layer MoS₂, serving as high- κ dielectrics and screening layers. Additionally, they demonstrated the potential of Bi₂SiO₅ as a gate dielectric in MoS₂ field-effect transistors. These transistors exhibited strong gate control over carrier densities and significant enhancements in mobility. Operating at a low gate voltage of 0.5 V, they demonstrated large Ion/Ioff ratios, minimal hysteresis, and low drain-induced barrier lowering (DIBL). Overall, this work holds promise for future electronics based on 2D materials and is expected to have a significant impact on the community. However, there are still important issues that need to be addressed before fully embracing this work. The following comments provide further insight into these matters.

1. As a material synthesis report, it is a pity that the growth mechanism is not discussed in the text. I suggest authors put more details and discuss how the reaction goes in CVD and what causes the temperature-dependent seeding process for in-plane and out-of-plane crystal growth.

2. I noticed that the dielectric constant of Bi₂SiO₅ layers prepared by the proposed method is way below the other reported value (Angew. Chem. Int. Ed., 52: 8088-8092, 2013 and J Mater Sci 56, 8415-8426, 2021). I suppose the dead-layer effect should not lead to such large degradation in single crystal layers. Can authors explain this phenomenon? And is it possible to improve?

3. Low DIBL occurs in such long-channel devices is not surprising. The performance of short-channel transistors is generally used to evaluate the gate controllability over the channel. I recommend providing some short channel performance to prove the superiority of Bi₂SiO₅ dielectric.

4. Please report your devices' area (Wch, Lch) in all Figures and give all currents as A/um or A/cm⁻² for better comparability.

5. Suggest authors normalize the hysteresis (mV/MV cm⁻¹) shown in your devices for different sweep rates and bias ranges and compare them to literature values.

Reviewer #1 (Remarks to the Author):

In this study, Chen et al reported high-quality ultrahigh-k Bi_2SiO_5 and constructed high-performance devices. Overall, I think the work could be impactful. But, I am not able to recommend a publication, unless the authors address the following issue.

Authors' response

We highly appreciate the positive evaluations and constructive suggestions on our work. Following the reviewer's suggestion, we paid particular attention on our instrumental noise level and the way of data presentation. For details, please see the point-by-point response below.

1. Bi_2SiO_5 is the central material of the study. Hence, it deserves a thorough introduction on its advantages and uniqueness, especially its theoretical and existing experimental results on dielectric constant, bandgap, and dielectric strength, to justify the application of this material.

Authors' response

Thanks for the reviewer's constructive suggestions. In our revision, we have added several descriptive sentences in the introduction part to strengthen the basic information of Bi_2SiO_5 .

2. All figures are very blurry, and most of the font sizes are very small. Many labels and values are impossible to read.

Authors' response

Thanks for the reviewer's kindness to point out the issue of low resolution and small font sizes in our figures. In fact, all figures in the word format are of high definition (600 dpi), but become blurry when converted into PDF format on line. In our revision, we re-edit the labels and values in adobe illustrator to make our figures more readable.

3. Figure 3b is noisy. Hence, the corresponding calculations of carrier density and dielectric constant in Fig. 3c should have a noticeable error bar and variation, respectively.

Authors' response

Thanks for the reviewer's professional suggestion. In our revision, based on the standard deviations of the fitted slopes in R_{xy} - B curves, the error bars of V_g -dependent carrier density were added (Fig. R1). As a result, the dielectric constant can be calculated as 36.00 ± 0.01 .

Fig. R1 | The extracted V_g -dependent sheet carrier density (n_{2D}), in which an ultrahigh

dielectric constant (ϵ_r) of 36.00 ± 0.01 was derived by linear fitting. The Error bars are based on the standard deviations of slope fitting on R_{xy} - B curves.

4. The author should present calibration on and raw data of capacitance measurement. In consideration of the small size, the capacitance measurement presented in Fig. 2 is generally very challenging and often produce large variations and errors. Hence, in order to reliably present the results, the author needs to present their protocol of offset calibration of the equipment, calibration of equipment noise level, effective area estimation, and dielectric constant calculation, at least in SI.

Authors’ response

Thanks for the reviewer’s professional comments and suggestions. Just as pointed out by the reviewer, we should take particular attention on the equipment noise/offset level and detection limit when perform C - V measurements on a capacitor with ultrasmall capacitances. In fact, very strict and standard calibration processes were operated in all our C - V measurements. Specifically, we carefully measured the open-load capacitance to evaluate the total noise/offset level caused by the LCR meter, cables and probe stations. As shown in Fig. R2a, our instrument has a capacitance offset of $<1.5 \times 10^{-14}$ F and capacitance noise of $<0.9 \times 10^{-14}$ F within the drive frequency from 100 Hz to 1 MHz. Comparatively, the absolute capacitance value of our Bi_2SiO_5 -based MIM device is about 1.1×10^{-12} F (Fig. R2b), which is 2~3 orders higher than the equipment noise/offset level. Besides, the effective area (A) of MIM device (namely the overlap area between top and bottom electrodes) is estimated as $95 \mu\text{m}^2$. The thickness of the Bi_2SiO_5 is measured as 25.6 nm by AFM. Therefore, the dielectric constant is extracted as 32.4 based on the equation of

$$C = \frac{A\epsilon_0\epsilon_{eff}}{d}$$

Fig. R2 | a The open-load C - V measurements within the drive frequency from 100 Hz to 1 MHz. **b** Raw data of the C - V measurements on a 25.6-nm-thick Bi_2SiO_5 nanoflake with a common MIM device configuration, where the thick graphite and In/Au metals serve as the bottom and top electrodes (inset), respectively.

In our revision, Fig. R2 is added in the Supporting information as Supplementary Fig. 12, and several descriptive sentences were also added to demonstrate our instrument noise/offset level and dielectric constant estimation process.

5. The breakdown voltage measured by AFM is local and not the same as the global one. The author shall admit to avoiding confusing the readers.

Authors' response

Yes, we totally agree that the *breakdown voltage measured by AFM is local and not the same as the global one*. In our revision, we clearly point it out in the main text that "*C-AFM is a very local and microscopic tool to measure the breakdown voltage of a dielectric insulator, which may be not the same with the global one determined by MIM device*".

6. For the FET measurement, if the SS is 62 mV/decade, the author cannot label 60 mV/decade in the figure. This will confuse the readers.

Authors' response

Thanks for the reviewer's professional suggestion. In our revision, we have deleted the label of 60 mV/decade in all our Figures to avoid possible misleading.

7. The author was not clear about which measurement was done using Keithley 2400 and PPMS. What equipment and type of cable did the author use to measure some of the figures, such as Figure 3b, 3e, 4c, 5b, 5e, and 5h? Keithley2400 and PPMS cable generally have high noise levels.

Authors' response

Thanks a lot for the reviewer to raise the professional question about our instrument's noise levels. Definitely, it is well known that performing reliable electrical measurements in the level of $\sim 10^{-13} \sim 10^{-14}$ A usually needs careful selection of cables and probe stations with excellent electromagnetic screening. As a result, high-quality tri-axis cables and electromagnetic shielding cover are indispensable during our FET measurements, whose off-state current is on the order of $\sim 10^{-13} \sim 10^{-14}$ A. As shown in Fig. R3a, our instruments including the semiconductor parameter analyzer, tri-axis cables and probe stations have a noise level of $\sim 1 \cdot 10^{-13}$ A within the voltage of ± 2 V, which is confirmed by the open-load *I-V* measurements. Nevertheless, just as pointed out by the reviewer, the electromagnetic shielding effect in our PPMS- Keithley source meter is not as good as our probe station, whose noise level is $\sim 10^{-10}$ A within the voltage of ± 2 V (Fig. R3b). However, it is still good enough to perform the Hall measurements and 4-probe transfer measurements to extract the mobility from the linear on-state part.

Fig. R3 | **a** Open-load *I-V* measurements on our instruments composed of semiconductor parameter analyzer, tri-axis cables and probe station. **b** Open-load *I-V* measurements on our PPMS-Keithley 2400 system.

Specifically, the gated Hall data (Fig. 3b), gated R_{xx} - T curves (Fig. 3d) and temperature-dependent 4-probe transfer curves (Fig. 4c) were measured on the PMMS system equipped with Keithley 2400, 6221 source meter and 2182 nanovoltmeter. The room-temperature two-probe transfer curves and output curves of Bi_2SiO_5 -based FETs (Fig. 3e, 3f, 4b, 5b, 5c, 5e and 5h) were measured on our commercial semiconductor parameter analyzer and probe stations.

In our revision, those above-mentioned important details were added in the experimental section.

8. Did the author normalize the curves or shift the curves vertically or horizontally?

Authors' response

Thanks for the reviewer's professional question. No normalization or shift was performed at all in all our measured electrical data.

Reviewer #2 (Remarks to the Author):

In this work, the authors report that ultrathin Bi_2SiO_5 crystals (down to monolayer) grown by chemical vapor deposition (CVD) can serve as an excellent gate dielectric for 2D semiconductors, showing an ultrahigh dielectric constant (>30), large band gap (~ 3.8 eV) and large breakdown field strength (9.4 MV/cm). Although the material is novel, the results of application in 2D dielectrics are far from satisfactory. I don't recommend to publish it in Nat. Comm.

Authors' response

We highly appreciate the reviewer's constructive suggestions and professional questions on our work, which really help us to improve the quality of our work. We should admit the imperfections in several of our electrical data in our last version, and thus new sets of experimental results were added. Besides, following the reviewer's suggestion, we adopted a more professional way to present our data in our revision. For details, please see the point-by-point response below.

1. It is well known that band offsets between semiconductors and dielectrics are important and necessary parameters in material and device design. For practical use, both the valence and conduction band offset (VBO and CBO) should be sufficiently large (> 1.0 eV) to block tunneling of electrons or holes from semiconductors. The bandgap of dielectrics usually needs to be larger than 5 eV. Therefore, claiming wide-gap insulator is not correct.

Authors' response

Thanks for the reviewer's kindness to point out the lack of rigor in our expression. In our revision, we have removed the phrase of "wide-gap insulator" throughout the manuscript.

2. There is a trade-off between K value and the bandgap condition, which requires a reasonably large band gap [Materials Science and Engineering R 88 (2015) 1–41]. Compared to the common dielectrics, the bandgap of this material is narrow, but its breakdown field is comparable to that of SiO_2 (the bandgap of SiO_2 is 9 eV). What is the mechanism behind it? The C-AFM measurement is reliable? In semiconductor area, MIM structure are usually used to evaluate the breakdown field.

Authors' response

Thanks for the reviewer's professional questions. Normally, as pointed out by the reviewer, a dielectric insulator with a high breakdown field usually needs a large band gap to prevent possible charge tunneling. However, the breakdown field of a dielectric insulator also depends on the quality of the crystals. In theory, the perfect single-crystalline dielectrics has a higher breakdown field (E_{bd}) than those polycrystalline or amorphous ones, since the electric breakdown usually occurs at the grain boundaries and defects. In our case, the CVD-grown Bi_2SiO_5 is single-crystalline with a band gap of ~ 3.8 eV, which is similar to the case of Bi_2SeO_5 (3.9 eV). The latest *Nature Materials* paper (DOI: 10.1038/s41563-023-01502-7) reported the E_{bd} of exfoliated Bi_2SeO_5 single-crystalline nanoflakes can be as high as 10~30 MV/cm by MIM measurements. Comparatively, the E_{bd} value of Bi_2SiO_5 single crystal determined as 7.2 to 9.4 MV/cm by C-AFM still locates in a reasonable range.

On the other hand, C-AFM is a well-known and widely-used local tool to measure the breakdown voltage of a dielectric insulator, which may be not the same with the global one determined by MIM device. To this end, we evaluated the E_{bd} of Bi_2SiO_5 with different thicknesses based on the MIM device. As shown in Fig. R4, the Bi_2SiO_5 -based device has an average E_{bd} value of 3~5 MV/cm. In this case, the extracted E_{bd} value of Bi_2SiO_5 still obeys the empirical rule mentioned by the reviewer, since the SiO_2 has a typical E_{bd} of ~ 10 MV/cm.

Fig. R4 | The leakage current as a function of applied voltage for Bi_2SiO_5 nanoflakes with various thicknesses from 10.1 to 21.4 nm.

In our revision, Fig. R4 was added in the supporting information as Supplementary Fig. 13, and we clearly point out in the main text that "C-AFM is a local and microscopic tool to measure the breakdown voltage of a dielectric insulator, which may be not the same with the global one determined by *MIM device*". Besides, we do not emphasize the breakdown strength of 9.4 MV measured by C-AFM in the abstract.

3. For dielectrics in 2D materials, the key parameters of transistors are mobility, drain current of the short channel, and EOT scaling. This paper did not focus on these important device parameters, which need to be improved to determine the potential of this material compared to BN and $\text{Bi}_2\text{O}_5\text{Se}$.

Authors' response

We appreciate the reviewer's professional questions. Yes, we totally agree that EOT scaling and

the short-channel FETs are both key parameters to check the potential of Bi_2SiO_5 as gate dielectrics. Following the reviewer's suggestion, we have performed extra experiments. Please see below for details.

Regarding the EOT scaling, we should emphasize that the thickness of in-plane grown Bi_2SiO_5 can be as thin as 3.9 nm, and its EOT can be as small as ~ 0.5 nm when a dielectric constant of ~ 30 is adopted. However, it is quite challenging to transfer ultrathin in-plane grown Bi_2SiO_5 crystals on mica to other substrates or integrate with other 2D semiconductors. As a result, we more emphasized the EOT values of vertically grown Bi_2SiO_5 that we have successfully fabricated the Bi_2SiO_5 -gated FET devices. The thinnest vertically grown Bi_2SiO_5 sample has a thickness of 7.5 nm, whose EOT equals to ~ 1 nm when its dielectric constant is supposed to be 30. In fact, we have tried to fabricate MoS_2 FETs with 7.5-nm-thick Bi_2SiO_5 as gate insulator, but wrinkles are inevitable during the device fabrication process. With great efforts, we only achieved the device fabrication of MoS_2 FETs with ~ 10 -nm-thick Bi_2SiO_5 as gate insulator, whose EOT is as small as 1.3. It is worth noting that the feature of easy transfer in vertically grown Bi_2SiO_5 greatly facilitates its vdWs integration with other 2D materials. Thanks to the high dielectric constant, the capacitance for a ~ 20 -nm-thick Bi_2SiO_5 can reach $\sim 1.5 \mu\text{F}/\text{cm}^2$, and the doped carrier density can be as high as $1.83 \times 10^{13} \text{ cm}^{-2}$ by applying a V_g of 2 V (Fig. 3 of main text). Consequently, we more emphasized its potential of vdWs integration as the single-crystalline vdWs substrate/dielectric to fabricate functional devices.

To check whether Bi_2SiO_5 can act as excellent gate dielectrics in short-channel FETs, we fabricate both back-gate and top-gate MoS_2 FETs with short channel lengths, as demonstrated in Figs. R5-7. As we know, fabricating short-channel device is quite technically challenging, which needs expensive lithography/evaporation facilities and well-trained operators. To be honest, our group is not skilled at this. However, with great efforts, we have successfully scaled the channel length of MoS_2 FET with graphene as the back-gate electrode to 100 nm (Fig. R5) and even 30 nm (Fig. R6). Notably, we adopted the previously reported two-step EBL method [*Science* 355, 271-276 (2017)] to separately deposit the source and drain electrodes, which is beneficial for the lift off of metal electrodes. As shown in Fig. R5, the Bi_2SiO_5 -gated MoS_2 FET with a 100 nm channel length can be effectively switched on and off, exhibiting an on/off ratio of $>10^3$, SS value of ~ 74 mV/decade, and DIBL value of ~ 30 mV/V. Furthermore, when the channel length is further scaled down to 30 nm, the Bi_2SiO_5 -gated MoS_2 FET still works well with a large on/off ratio. The relatively larger SS value of ~ 170 mV/decade and DIBL value of ~ 65 mV/V indicate substantial room space for device optimization. Besides, the On-state current of MoS_2 -based short-channel FET can be as high as $\sim 35 \mu\text{A}/\mu\text{m}$, which is believed to be further improved by precise contact engineering.

Fig. R5 | A back-gate MoS₂ short-channel FET ($L_{CH} = 100 \text{ nm}$) with graphene and Bi₂SiO₅ as the back-gate electrode and dielectrics. **a OM image of the as-fabricated MoS₂ FET with a channel length of 100 nm. **b** The corresponding AFM image, showing the channel length of ~ 100 nm. **c** Transfer characteristics of the device at $V_{ds} = 0.05 \text{ V}$ (red) and 0.50 V (blue), showing a large on/off ratio of $>10^8$, small SS of 74 mV/decade and low DIBL value of 30 mV/V. **d** The corresponding output curves under different V_g . From top to bottom, V_g varies from 3.0 to 0.0 V with a step of 0.3 V.**

Fig. R6 | Preliminary results of a back-gate MoS₂ short-channel FET ($L_{CH} = 30$ nm) with graphene and Bi₂SiO₅ as the back-gate electrode and dielectrics. **a Schematic of the device structure. **b** SEM image of the as-fabricated MoS₂ FET device, showing the channel length of ~ 30 nm. **c** Transfer characteristics of the device at $V_{ds} = 0.05$ V (red) and 0.20 V (blue), showing a large on/off ratio of $>10^8$. Its SS value is ~ 170 mV/decade, and DIBL value is 65 mV/V. **d** The corresponding output curves under different V_g . From top to bottom, V_g varies from 3.6 V to 0.0 V with a step of 0.2 V.**

One step further, we fabricated the short-channel MoS₂ FETs with Bi₂SiO₅ as the top-gate dielectrics and graphene as the contact electrodes. The channel length of the MoS₂ FETs was defined by the gap distance between two graphene electrodes (~ 180 nm), and the graphene gap was prepared by the EBL process and subsequent etching of O₂ plasma. As shown in Fig. R7, the top-gate MoS₂ FET with a channel length of ~ 180 nm exhibited a large on/off ratio of $>10^7$, small SS value of ~ 79 mV/decade and DIBL value of 22 mV/V. Our preliminary data of short-channel FETs further illustrated the potential of Bi₂SiO₅ as the gate dielectrics.

Fig. R7 | A top-gate MoS₂ short-channel FET with graphene as the contact electrodes and Bi₂SiO₅ as the gate dielectrics. **a Schematic of the device structure. **b, c** OM image (**b**) and AFM image (**c**) of the graphene electrodes with a gap distance of 180 nm, which was produced by EBL process and subsequent etching of O₂ plasma. **d** OM image of as-fabricated Bi₂SiO₅-gated MoS₂ short-channel FET with graphene as the contact electrodes. **e** Transfer curves of the device at $V_{ds} = 0.05$ V (red) and 0.50 V (blue), showing a small SS of 79 mV/decade and DIBL value of 22 mV/V. **f** The corresponding output curves under different V_g . From top to bottom, V_g varies from 2.0 to 0.0 V with a step of 0.2 V.**

In our revision, Figs. R5-7 were added in the supporting information as new Supplementary Figs. 16, 17, 21, and the above paragraphs were also added in the SI to describe the new short-channel data.

4. All the drain currents in this work should be normalized by channel width.

Authors' response

We thank the referee for his/her professional suggestion. In our revision, all the drain currents in this work are normalized by channel width.

5. In fig. 4, the MoS₂ devices on Bi₂SiO₅ and SiO₂ have different threshold voltages, indicating different carrier densities in fig. 4b. Therefore, to make a fair comparison, using $V_g - V_{th}$ is recommended.

Authors' response

We thank the very insightful comments and suggestions from the reviewer. As shown in Fig. R8, we re-plotted the 4-probe transfer curves by using the form of I_{ds} as a function of $V_g - V_{th}$.

Fig. R8 | I_{ds} as a function of $V_g - V_{th}$ in MoS₂ FETs measured at different temperatures (5~300 K) on Bi₂SiO₅ (left) and SiO₂ (right) substrates, respectively.

In our revision, the Fig. R8 is added in the SI as Supplementary Fig. 18.

6. In Fig. 5f, the shift of V_{th} is just ~ 0.05 V. However, in fig. 5e, the shift is actually ~ 0.2 V by changing V_{bg} if $I_{ds} = 0.1$ nA. Therefore, the slope is problematic.

Authors' response

Thanks for the reviewer's professional question based on his/her close-up view of the V_{BG} -varied transfer curves (Fig. 5e of main text), whose I_{ds} is shown on a logarithmic scale. However, to exact the correct V_{th} of the FET, we should present the $I_{ds} - V_g$ curve on a nonlogarithmic scale [Nat. Electron. 2, 563–571 (2019)]. Figure R9 showed the raw data of the transfer curves on a nonlogarithmic scale and the way how we extract the V_{th} . We should emphasize that the V_{th} value is mainly determined by the shift of the on-state part rather than the depletion-state part.

Fig. R9 | The threshold voltage V_{th} exacted by linear fitting the transfer curves on a

nonlogarithmic scale. a-f Transfer curves of the device (raw data of Fig. 5e in the main text) under different back-gate voltages V_{BG} from 5 to 0 V.

In our revision, the Fig. R9 was added in the SI as new supplementary Fig. 22, and a new reference paper was added.

7. In Fig. 5h, the performance of MoS₂ transistor decreases with decreasing the thickness of BiSiO. What is the reason? The V_g range is too small. Pushing it to 2V is reasonable in view of high breakdown field of BiSiO.

Authors' response

Thanks for the reviewer's professional questions. Yes, we should admit that Fig. 5h presented a sign that decreasing the thickness of Bi₂SiO₅ will decrease the on-state current of MoS₂ FET, which may originate from the contact issues existing in the MoS₂ FETs with the thin Bi₂SiO₅ as gate insulators.

To illuminate the possible confusion, we have fabricated new batches of MoS₂ FET devices with ultrathin Bi₂SiO₅ as the top-gate dielectrics. As shown in Fig. R10, whose gate voltage was pushed to 2 V as requested, the on-state current of the 10-nm-thick Bi₂SiO₅ gated MoS₂ FET also can be as high as 0.16 $\mu\text{A}/\mu\text{m}$. This value is very similar to the results of the thick Bi₂SiO₅ gated FETs (0.11 $\mu\text{A}/\mu\text{m}$, Fig. 5b of the main text). It is worth noting that the on-state current is greatly limited by the remaining ungated channel in our top-gate device configuration. Besides, as confirmed by the dual-sweep transfer curve in Fig. R10a, a nearly ideal SS value of ~ 62 mV/decade and ignorable gate hysteresis were also obtained in the 10-nm-thick Bi₂SiO₅ gated MoS₂ FET.

Fig. R10 | Another top-gate MoS₂ FET with 10-nm-thick Bi₂SiO₅ as gate dielectrics. a Dual-sweep transfer curves of the MoS₂/ Bi₂SiO₅ FET measured under different V_{ds} from 0.05 to 1 V, showing an ideal SS value of ~ 62 mV/decade and ignorable gate hysteresis. By pushing the gate voltage to 2 V, the I_{ds} saturated gradually. The OM image of the fabricated device is inserted in **a**. **b** Corresponding output curves of the device measured by varying the V_g from 2.0 to -1.0 V with a step of 0.2 V.

In our revision, the Fig. R10 was added as new supplementary Fig. 23 in the SI.

8. To evaluate the interface trap, the interface trap density D_{it} is a quantified parameter. Please provide this information.

Authors' response

We highly appreciate the reviewer's professional suggestion. The interface trap density D_{it} is extracted based on the following equation [see *Nature* 605, 262 (2022)]:

$$SS = \ln(10) \frac{k_B T}{q} \left(1 + \frac{q D_{it}}{C_{ox}} \right)$$

where k_B is Boltzmann constant, T is absolute temperature, q is the elementary charge, C_{ox} is the gate capacitance obtained from MOS capacitance measurements ($1.25 \times 10^{-2} F m^{-2}$). The extracted D_{it} is about $2.88 \times 10^{11} cm^{-2}/eV$.

In our revision, the D_{it} value and several descriptive sentences were added in the main text.

Reviewer #3 (Remarks to the Author):

Chen et al. achieved successful growth of ultrathin Bi_2SiO_5 crystals using chemical vapor deposition (CVD). These crystals possess advantageous properties, including a high dielectric constant, a large band gap, and high breakdown field strength. An important feature of the few-layer Bi_2SiO_5 dielectrics is their vertical growth on a mica substrate with weak interfacial interaction. This allows for easy transfer onto other substrates through mechanical pressing, facilitating ideal van der Waals integration with other 2D materials like few-layer MoS_2 , serving as high- κ dielectrics and screening layers. Additionally, they demonstrated the potential of Bi_2SiO_5 as a gate dielectric in MoS_2 field-effect transistors. These transistors exhibited strong gate control over carrier densities and significant enhancements in mobility. Operating at a low gate voltage of 0.5 V, they demonstrated large I_{on}/I_{off} ratios, minimal hysteresis, and low drain-induced barrier lowering (DIBL). Overall, this work holds promise for future electronics based on 2D materials and is expected to have a significant impact on the community. However, there are still important issues that need to be addressed before fully embracing this work. The following comments provide further insight into these matters.

Authors' response

We really appreciate the reviewer's positive evaluations and constructive suggestions on our work, which really help us to greatly improve the quality of our work. Following the reviewer's suggestion, we have done extra experiments. Please see below for our point-by-point response.

1. As a material synthesis report, it is a pity that the growth mechanism is not discussed in the text. I suggest authors put more details and discuss how the reaction goes in CVD and what causes the temperature-dependent seeding process for in-plane and out-of-plane crystal growth.

Authors' response

We highly appreciate that the reviewer raised a question about the CVD growth mechanism of Bi_2SiO_5 . If we understand correctly, the reviewer's question contains two parts: 1) how the reaction goes in the CVD growth of Bi_2SiO_5 ; 2) the possible reason for the in-plane and out-of-plane growth.

1) One step further to understand how the reaction goes in the CVD growth of Bi_2SiO_5 ? First, it's well-known that the CVD growth usually undergoes a very complex microscopic process, including thermal decomposition/volatilization, gas transport, and absorption-diffusion-desorption of the chemical precursors in the form of atomic clusters. Even so, we can still try to understand the CVD growth process from the viewpoint that the compounds are formed by the combination of key elements-containing precursors. Undoubtedly, if one wants to synthesize the Bi_2SiO_5 , it needs the Bi and Si precursors during the CVD growth process. Definitely, BiF_3 powder is the only possible compound to supply the Bi-containing precursor. However, our CVD setup has a much more complex environment for Si. For example, the quartz boat, quartz tube and even the mica substrate can act as the Si-precursor supplier. To figure out the dominated Si-supplier, we performed extra comparison experiments. As shown in Fig. R11, when the BiF_3 powders are directly placed in the quartz boat container, Bi_2SiO_5 nanosheets can be readily obtained no matter on mica substrate (Fig. R11a) or sapphire substrate (Fig. R11b-c). If the quartz boat was replaced by the corundum boat (Al_2O_3) with SiO_2 powders inside, we can still get the Bi_2SiO_5 phase (Fig. R11d). However, if no SiO_2 powders were placed in the corundum boat, we can't obtain the Bi_2SiO_5 phase at all, but a delta- Bi_2O_3 phase with a triangular like morphology was obtained instead (Fig. R11e), whose crystal structure was verified by the cross sectional TEM characterizations (Fig. R11f-h). The lattice spacings of 0.33 nm, 0.28 nm and 0.20 nm match well with the (111), (002) and (220) planes in delta- Bi_2O_3 . It suggests that the quartz boat may be the dominated parameter to supply Si and react with BiF_3 to synthesize the Bi_2SiO_5 phase.

Fig. R11 | Comparison experiments conducted to figure out which part of SiO_2 dominates the CVD growth to synthesize the phase of Bi_2SiO_5 . **a** A typical OM image of Bi_2SiO_5 nanosheets grown on mica substrate when the BiF_3 powders are directly placed in the quartz boat container. **b** A typical SEM image of Bi_2SiO_5 nanosheets grown on sapphire substrate when the BiF_3 powders are directly placed in the quartz boat container. **c** The corresponding XRD pattern of as-synthesized Bi_2SiO_5 on the sapphire substrate. **d** A typical OM image of Bi_2SiO_5 nanosheets grown on mica substrate when mixing the SiO_2 powders with the BiF_3 powders in the corundum boat. **e** A typical OM image of the

delta-Bi₂O₃ phase with a triangular like morphology when the BiF₃ powders are placed in the corundum boat container. **f, g** Cross-sectional TEM data of the triangular-like crystals imaging along the zone axes of [11 $\bar{2}$, **f**], and [10 $\bar{1}$, **g**], respectively. **h** The corresponding EDS elemental mappings for Bi, O and Si.

2) The possible reason for the in-plane and out-of-plane growth.

First, we should emphasize that we can faithfully and repeatedly observe the phenomenon that lowering the growth temperature can regulate the CVD growth mode from in-plane to out-of-plane. As shown in Fig. R12, we repeated each of CVD growth for in-plane and out-of-plane growth for 3 times, showing the very similar growth results. To this end, the question comes into why temperature matters for the growth mode of Bi₂SiO₅. Typically, the most direct way to address this issue is to calculate the nucleation energy barriers between different lattice planes of Bi₂SiO₅ and mica. However, this kind of calculation is quite challenging since we should know the exact form of the precursor clusters on mica or Bi₂SiO₅ surfaces during CVD growth. Here, we can do a qualitative analysis on this phenomenon. We should point out that altering the growth mode or aspect ratio of a nanomaterial by changing the synthetic temperature is a widely used method in other material systems. For example, Hong, C. Y. *et al.* reported the very similar temperature-induced vertical growth results in Bi₂O₂Se [ACS Nano 14, 16803-16812 (2020)], whose crystal structure is similar to Bi₂SiO₅. As we know, the species and relative partial pressure of the precursors would be different at different temperatures, which will also affect the nucleation energy barrier. Generally speaking, the vertical growth has a chemical bond-like interfacial interaction, which may lead to a lower nucleation energy barrier for vertical growth at low temperature. With growth temperature increasing, the adatoms (or precursor clusters) on the mica substrate are excited with higher kinetic energy to achieve a longer diffusion distance, and then the in-plane growth of Bi₂SiO₅ could be boosted.

Fig. R12 | Repeated CVD growth results of Bi₂SiO₅ with different growth modes. Each of them

was repeated for 3 times, showing very similar results. **a-c** OM images of in-plane grown Bi₂SiO₅ nanosheets on mica at a relatively high temperature of 1023 K. **d-f** SEM images of vertically grown Bi₂SiO₅ nanosheets on mica at a relatively low temperature of 923 K.

In our revision, we added a whole part to discuss the possible CVD process of Bi₂SiO₅ and reasons for the in-plane and out-of-plane growth. Besides, the Figs. R11, 12 were also added as new supplementary Figs. 7, 8 in SI.

2. I noticed that the dielectric constant of Bi₂SiO₅ layers prepared by the proposed method is way below the other reported value (Angew. Chem. Int. Ed., 52: 8088-8092, 2013 and J. Mater. Sci. 56, 8415–8426, 2021). I suppose the dead-layer effect should not lead to such large degradation in single crystal layers. Can authors explain this phenomenon? And is it possible to improve?

Authors' response

Thanks for the reviewer's professional question. First of all, let me make it clear that the dielectric constant of a layered material is highly related to the exact crystal planes. In other words, different crystal planes usually have different dielectric constants, and this is also the case in Bi₂SiO₅ [Angew. Chem. Int. Ed. 52, 8088-8092 (2013)]. For the case of polycrystalline powders, the dielectric constant measured is an average value related with different crystal planes. In our work, the CVD-grown Bi₂SiO₅ is a single crystal, whose out of plane is parallel to its *a*-axis. As a result, all our *C-V* and gated Hall data gave the out-of-plane dielectric constant of Bi₂SiO₅.

After careful searching the literature, we found that in-plane dielectric constant of Bi₂SiO₅ is higher than its out-of-plane value. To this end, if one can achieve the growth of Bi₂SiO₅ single crystals with other preferred lattice directions, it is possible to further improve its dielectric constant. Besides, its out-of-plane dielectric constant was reported in the range of ~30 to 80 in different works. For example, Taniguchi, H. *et al.* [Angew. Chem. Int. Ed. 52, 8088-8092 (2013)] reported the out-of-plane dielectric constant of Bi₂SiO₅ single crystal can be as high as ~80 at room temperature, but Kijima, T. *et al.* [Jpn. J. Appl. Phys. 37, 5171-5173 (1998)] reported that out-of-plane dielectric of Bi₂SiO₅ film on SrTiO₃ is about 30. Several other works of Bi₂SiO₅ powders, such as Sakamoto, K. *et al.* reported the dielectric constant of polycrystalline Bi₂SiO₅ is about 60~75 at room temperature [J. Mater. Sci. 56, 8415–8426, (2021)]. Based on previously reported results, we believe the out-of-plane dielectric constant of Bi₂SiO₅ extracted as ~30 in our work is reasonable.

3. Low DIBL occurs in such long-channel devices is not surprising. The performance of short-channel transistors is generally used to evaluate the gate controllability over the channel. I recommend providing some short channel performance to prove the superiority of Bi₂SiO₅ dielectric.

Authors' response

Thanks for the reviewer's constructive suggestion. To check whether Bi₂SiO₅ can act as excellent gate dielectrics in short-channel FETs, we fabricate both back-gate and top-gate MoS₂ FETs with short channel lengths, as demonstrated in Fig. R13-15. As we all know, fabricating short-channel device is quite technically challenging, which needs expensive lithography/evaporation facilities and well-trained operators. To be honest, our group is not skilled at this. However, with great

efforts, we have successfully scaled the channel length of MoS₂ FET with graphene as the back-gate electrode to 100 nm (Fig. R13) and even 30 nm (Fig. R14). Notably, we adopted the previously reported two-step EBL method [*Science*. 355, 271-276 (2017)] to separately deposit the source and drain electrodes, which is beneficial for the lift off of metal electrodes. As shown in Fig. R13, the Bi₂SiO₅-gated MoS₂ FET with a 100 nm channel length can be effectively switched on and off, exhibiting an on/off ratio of >10⁸, SS value of ~74 mV/decade, and DIBL value of ~30 mV/V. Furthermore, when the channel length is further scaled down to 30 nm, the Bi₂SiO₅-gated MoS₂ FET still works well with a large on/off ratio. The relatively larger SS value of ~160 mV/decade and DIBL value of ~65 mV/V indicate substantial room space for device optimization. Besides, the On-state current of MoS₂-based short channel length FET can be as high as ~35 μA/μm, which is believed to be further improved by precise contact engineering.

Fig. R13 | A back-gate MoS₂ short-channel FET ($L_{CH} = 100$ nm) with graphene and Bi₂SiO₅ as the back-gate electrode and dielectrics. **a OM image of the as-fabricated MoS₂ FET with a channel length of 100 nm. **b** The corresponding AFM image, showing the channel length of ~ 100 nm. **c** Transfer characteristics of the device at $V_{ds} = 0.05$ V (red) and 0.50 V (blue), showing a large on/off ratio of >10⁸, small SS of 74 mV/decade and low DIBL value of 30 mV/V. **d** The corresponding output curves under different V_g . From top to bottom, V_g varies from 3.0 to 0.0 V with a step of 0.3 V.**

Fig. R14 | Preliminary results of a back-gate MoS₂ short-channel FET ($L_{CH} = 30$ nm) with graphene and Bi₂SiO₅ as the back-gate electrode and dielectrics. **a Schematic of the device structure. **b** SEM image of the as-fabricated MoS₂ FET device, showing the channel length of ~ 30 nm. **c** Transfer characteristics of the device at $V_{ds} = 0.05$ V (red) and 0.20 V (blue), showing a large on/off ratio of $>10^8$. Its SS value is ~ 170 mV/decade, and DIBL value is 65 mV/V. **d** The corresponding output curves under different V_g . From top to bottom, V_g varies from 3.6 V to 0.0 V with a step of 0.2 V.**

One step further, we fabricated the short-channel MoS₂ FETs with Bi₂SiO₅ as the top-gate dielectrics and graphene as the contact electrodes. The channel length of the MoS₂ FETs was defined by the gap distance between two graphene electrodes (~ 180 nm), and the graphene gap was prepared by the EBL process and subsequent etching of O₂ plasma. As shown in Fig. R15, the top-gate MoS₂ FET with a channel length of ~ 180 nm exhibited a large on/off ratio of $>10^7$, small SS value of ~ 79 mV/decade and DIBL value of 22 mV/V. Our preliminary data of short-channel FETs further illustrated the potential of Bi₂SiO₅ as the gate dielectrics.

Fig. R15 | A top-gate MoS₂ short-channel FET with graphene as the contact electrodes and Bi₂SiO₅ as the gate dielectrics. a Schematic of the device structure. **b, c** OM image (**b**) and AFM image (**c**) of the graphene electrodes with a gap distance of 180 nm, which was produced by EBL process and subsequent etching of O₂ plasma. **d** OM image of as-fabricated Bi₂SiO₅-gated MoS₂ short-channel FET with graphene as the contact electrodes. **e** Transfer curves of the device at $V_{ds} = 0.05$ V (red) and 0.50 V (blue), showing a small SS of 79 mV/decade and DIBL value of 22 mV/V. **f** The corresponding output curves under different V_g . From top to bottom, V_g varies from 2.0 to 0.0 V with a step of 0.2 V.

In our revision, Figs. R13-15 were added in the supporting information as new Supplementary Figs. 16, 17, 21, and the above paragraphs were also added in the SI to describe the new short-channel data.

4. Please report your devices' area (W_{ch} , L_{ch}) in all Figures and give all currents as A/μm or A/cm² for better comparability.

Authors' response

We thank the referee for his/her professional suggestion. In our revision, the width and length of the device are reported, and all the drain currents are normalized by channel width as A/μm.

5. Suggest authors normalize the hysteresis (mV/MV cm⁻¹) shown in your devices for different sweep rates and bias ranges and compare them to literature values.

Authors' response

Thanks for the reviewer's professional suggestion. To address the question raised by the reviewer,

we fabricated new top-gate MoS₂ FETs with 10-nm-thick Bi₂SiO₅ as the gate dielectric. As shown in Fig. R16, the Bi₂SiO₅-gated MoS₂ FET showed a near ideal SS value of 62 mV/decade and ignorable hysteresis for different sweep rates (Fig. R17a-c) and bias ranges (Fig. R17d-f).

Subsequently, we normalized the hysteresis for different sweep rates and bias ranges. The Norm. ΔV_H is calculated by using $\Delta V_H/E_g$, where ΔV_H is the gate hysteresis and E_g is the gate electric field. The gate electric field $E_g=V_g/d_{ox}$, where the d_{ox} is the thickness of oxide films and the V_g is the gate voltage. [*Nat. Electron.* 2, 230 (2019); *Nat. Commun.* 14, 2340 (2023)]. As shown in Fig. R17c and f, the Norm. ΔV_H is in the range of 9~37 mV/MV cm⁻¹. Furthermore, as shown in Fig. R18, we compared our lowest Norm. ΔV_H with other literatures [*Nat. Commun.* 14, 2340 (2023); *Nature* 605, 262 (2022); *Nat. Electron.* 5, 643 (2022); *Nat. Electron.* 2, 563 (2019); *AIP Adv.* 5, 057102 (2015); *IEEE Electr. Device L.* 38, 1763 (2017); *Appl. Phys. Express* 9, 095202 (2016)]. The ultralow Norm. ΔV_H further verified the high-quality interface obtained in our device.

In our revision, Figs. R16-18 were added in the supporting information as Supplementary Figs. 24, 25.

Fig. R16 | Another top-gate MoS₂ FET with 10-nm-thick Bi₂SiO₅ as gate dielectrics. a Dual-sweep transfer curves of the MoS₂/ Bi₂SiO₅ FET measured under different V_{ds} from 0.05 to 1 V, showing an ideal SS value of ~ 62 mV/decade and ignorable gate hysteresis. The sweep rate of the V_g was kept constant as 0.1 V s^{-1} . By pushing the gate voltage to 2 V, the I_{ds} saturated gradually. The OM image of the fabricated device is inserted in **a**. **b** Corresponding output curves of the device measured by varying the V_g from 2.0 to -1.0 V with a step of 0.2 V.

Fig. R17 | a, b Dual-sweep transfer characteristics of a top-gate MoS₂ FET with the 10-nm-thick Bi₂SiO₅ (the same device with Fig. R16) under different sweep rates. The sweep speeds varied from 0.25 to 0.025 V s⁻¹. **c** Extracted hysteresis and normalized hysteresis at different sweep rates. **d, e** dual-sweep transfer curves measured under different V_g ranges. **f** Extracted hysteresis and Normalized hysteresis at different V_g ranges.

Fig. R18. Comparison of normalized hysteresis in our device with other literatures.

REVIEWERS' COMMENTS

Reviewer #1 (Remarks to the Author):

In the revision, the authors did address my questions nicely. But, the inset of Fig. 3e is still impossible to read, and its font shall be enlarged. Other than this, I am ok to recommend a publication.

Reviewer #2 (Remarks to the Author):

The authors have addressed my major concerns. However, for fig. 5h, the authors believe that decreasing the thickness of Bi₂SiO₅ will decrease the on-state current of MoS₂ FET due to the contact issues existing in the MoS₂ FETs with the thin Bi₂SiO₅ as gate insulators. There are not references to support it. It is well known that contact resistance is mainly related to mobility of channel materials and contact metal material. Therefore, I don't think the explanation in this paper is right. In general, dielectric thickness (10-60 nm) do not affect contact resistance and mobility. The advantages of Bi₂SiO₅ are high k with good SS and hysteresis. However, the thickness scaling decreases the performance of transistor, which will limit the potential of this material for use.

Reviewer #3 (Remarks to the Author):

I would like to thank the authors for their diligent response to my concerns about short-channel performance, dielectric evaluation, and growth mechanism. From my perspective, the papers can be basically accepted as is.

Reviewer #1 (Remarks to the Author):

In the revision, the authors did address my questions nicely. But, the inset of Fig. 3e is still impossible to read, and its font shall be enlarged. Other than this, I am ok to recommend a publication.

Authors' response

Thanks for the reviewer's positive evaluation on our last-round revision. All the suggestions and comments really help us to improve the quality of our work. In our revised version, we enlarged the font size of the inset of Fig. 3e.

Reviewer #2 (Remarks to the Author):

The authors have addressed my major concerns. However, for fig. 5h, the authors believe that decreasing the thickness of Bi_2SiO_5 will decrease the on-state current of MoS_2 FET due to the contact issues existing in the MoS_2 FETs with the thin Bi_2SiO_5 as gate insulators. There are no references to support it. It is well known that contact resistance is mainly related to mobility of channel materials and contact metal material. Therefore, I don't think the explanation in this paper is right. In general, dielectric thickness (10-60 nm) do not affect contact resistance and mobility. The advantages of Bi_2SiO_5 are high k with good SS and hysteresis. However, the thickness scaling decreases the performance of transistor, which will limit the potential of this material for use.

Authors' response

Thanks for the reviewer's positive evaluation and constructive suggestions on our work. Yes, just as pointed out by the reviewer, we totally agree that dielectric thick scaling from 60 to ~10 nm will not affect the contact resistance of the FET in theory. We should emphasize that the on-state current (I_{on}) of MoS_2 FETs with 10-nm-thick Bi_2SiO_5 as gate insulators varies from device to device. The relatively small I_{on} for the ~10-nm-thick Bi_2SiO_5 gated MoS_2 FET in fig. 5h is only obtained on a device which may occasionally has a very large contact resistance. With optimization of device fabrication process, the I_{on} of the ultrathin Bi_2SiO_5 gated MoS_2 FET can be greatly improved. As shown in Supplementary Fig. 23 and Fig. R1, the I_{on} of another ~10-nm-thick Bi_2SiO_5 gated MoS_2 FET can be as high as $0.16 \mu\text{A}/\mu\text{m}$, which is very similar to the value of the thick Bi_2SiO_5 -gated FETs ($0.11 \mu\text{A}/\mu\text{m}$, Fig. 5b). In other words, no apparent performance degradation of the transistors was observed upon thickness scaling of Bi_2SiO_5 .

In our revision, to avoid possible misleading, we added a short sentence and modified the sentence structures to emphasize the data of supplementary Fig. 23.

Fig. R1. Transfer curves of another 10-nm-thick Bi_2SiO_5 -gated MoS_2 FET (red) compared to the data obtained on a 22.9-nm-thick Bi_2SiO_5 gated FET (blue, Fig. 5b of main text). The V_{ds} is 1 V.

Reviewer #3 (Remarks to the Author):

I would like to thank the authors for their diligent response to my concerns about short-channel performance, dielectric evaluation, and growth mechanism. From my perspective, the papers can be basically accepted as is.

Authors' response

We appreciate the reviewer for his/her positive evaluation and constructive suggestions on our work, which really help us to improve the quality of our work.